# A few Ascomycota taxa dominate soil fungal communities worldwide

Eleonora Egidi[1], Manuel Delgado-Baquerizo[1,2,3], Jonathan M. Plett[1], Juntao Wang[1,4], David J. Eldridge[5], Richard D. Bardgett [6], Fernando T. Maestre [3,7] & Brajesh K. Singh [1,8]

Despite having key functions in terrestrial ecosystems, information on the dominant soil fungi and their ecological preferences at the global scale is lacking. To fill this knowledge gap, we surveyed 235 soils from across the globe. Our findings indicate that 83 phylotypes (<0.1% of the retrieved fungi), mostly belonging to wind dispersed, generalist Ascomycota, dominate soils globally. We identify patterns and ecological drivers of dominant soil fungal taxa occurrence, and present a map of their distribution in soils worldwide. Whole-genome comparisons with less dominant, generalist fungi point at a significantly higher number of genes related to stress-tolerance and resource uptake in the dominant fungi, suggesting that they might be better in colonising a wide range of environments. Our findings constitute a major advance in our understanding of the ecology of fungi, and have implications for the development of strategies to preserve them and the ecosystem functions they provide.

[1] Hawkesbury Institute for the Environment, Western Sydney University, Richmond, NSW 2753, Australia. [2] Cooperative Institute for Research in Environmental Sciences, University of Colorado, Boulder, CO 80309, USA. [3] Departamento de Biología y Geología, Física y Química Inorgánica, Escuela Superior de Ciencias Experimentales y Tecnología. Universidad Rey Juan Carlos, c/Tulipán s/n, 28933 Móstoles, Spain. [4] State Key Laboratory of Urban and Regional Ecology, Research Center for Eco-Environmental Sciences, Chinese Academy of Sciences, Beijing 100085, China. [5] Centre for Ecosystem Science, School of Biological, Earth and Environmental Sciences, University of New South Wales, Sydney, NSW 2052, Australia. [6] School of Earth and Environmental Sciences, Michael Smith Building, The University of Manchester, Oxford Road, Manchester M13 9PT, UK. [7] Departamento de Ecología and Instituto Multidisciplinar para el Estudio del Medio "Ramon Margalef", Universidad de Alicante, Alicante, Spain. [8] Global Centre for Land-Based Innovation, Western Sydney University, Penrith South DC, NSW 2751, Australia. Correspondence and requests for materials should be addressed to E.E. (email: e.egidi@westernsydney.edu.au) or to M.D.-B. (email: mandelbaq@gmail.com) or to B.K.S. (email: b.singh@westernsydney.edu.au)

Soil fungi are among the most abundant and diverse taxonomic groups on Earth[1], are pathogens and mutualistic symbionts of both plants and animals[2], and play essential roles in ecosystems, such as pedogenesis, nutrient cycling, and disease suppression[3]. Given their prominent contribution to key terrestrial processes, information about their ecology and biogeography is of primary importance to prioritise ecosystem-level conservation and management efforts.

Like many other microbes, high-throughput sequencing technologies have significantly impacted our perception of fungal diversity and ecology, contributing to uncover the role of environmental attributes in shaping richness and distribution of soil fungal communities at both local and global scales. At wide scales, many fungal taxa appear to be limited by a combination of physical barriers, abiotic features, host occurrence and genetic restrictions on adaptation[4], and their distribution to be restricted to local or regional areas. For example, phylogeographic studies on medically relevant or economically important individual species of putative ubiquitous fungi, such as crop and human pathogens, have often retrieved narrowly distributed cryptic phylogenetic species[5–8]. More recently, many meta-barcoding efforts consistently reported significant distance-decay relationships and scarce phylotype-level community compositional overlap amongst communities of soil fungi at large scales[9–11].

While accumulating evidence suggests a strong spatial structuring of soil fungal communities across ecological gradients, indications of large-scale dispersal and ability for some fungi to dominate many environments also exist. For instance, some fungal phylotypes (e.g., members of the genera *Cladosporium*, *Toxicocladosporium*, and *Alternaria*) possess potentially widespread distributions, and may also be highly predominant in different ecosystems in terms of relative abundance[12,13]. Similarly, the connectedness of biogeographic regions of the world by shared fungal phylotypes indicates that a number of fungi can be detected in multiple continents and biomes[14,15]. Yet, a systematic and comprehensive assessment of diversity, identity, ecology, and distribution of those abundant and ubiquitous fungal taxa dominating the soil across the globe is still lacking, mainly due to limitations in the biomes and soil types covered by previous studies[14,16].

Identifying and characterising these cosmopolitan and abundant fungi represents a goal of theoretical and practical significance[17]. Interactions among dominant taxa are predicted to disproportionally affect community stability and functioning[18–20], particularly among natural microbial communities[21]. As such, determining which fungi are dominant in soils, the environmental variables that drive their abundance and distribution, and common mechanisms underlying dominance capabilities, constitute a major scientific advance. This knowledge can also help us to develop tools to predict how they may respond to ongoing environmental changes, ultimately leading to management strategies to improve fungi-mediated ecosystem functions and services.

To build novel insights into the identity, global distribution and ecology of dominant soil fungi, we identify the most ubiquitous fungal phylotypes using soils collected from 235 sites covering six continents and nine biomes of the word (Supplementary Fig. S1). We then characterise their distribution and habitat preferences in relation to edaphic and climatic parameters ("Methods"). Furthermore, we assess whether taxonomic relationships and life-styles can help explain the observed patterns of fungal dominance. Finally, we use information from fully sequenced genomes matching dominant fungal phylotypes identified in this study to determine whether genomic traits associated with characters such as competitive ability, stress resistance, nutrient acquisition, metabolism type and carbohydrate degradation are related to these patterns.

## Results

**Identification of dominant fungal phylotypes.** The sequencing resulted in 47,207 reads/564 phylotypes per sample (Supplementary Fig. 2A). We found that 83 fungal phylotypes were dominant in soils worldwide. These phylotypes represented <0.1 % of all the 24,137 sampled fungal phylotypes, while contributing to a fifth (~18%) of the total reads identified as fungi (Fig. 1). We further

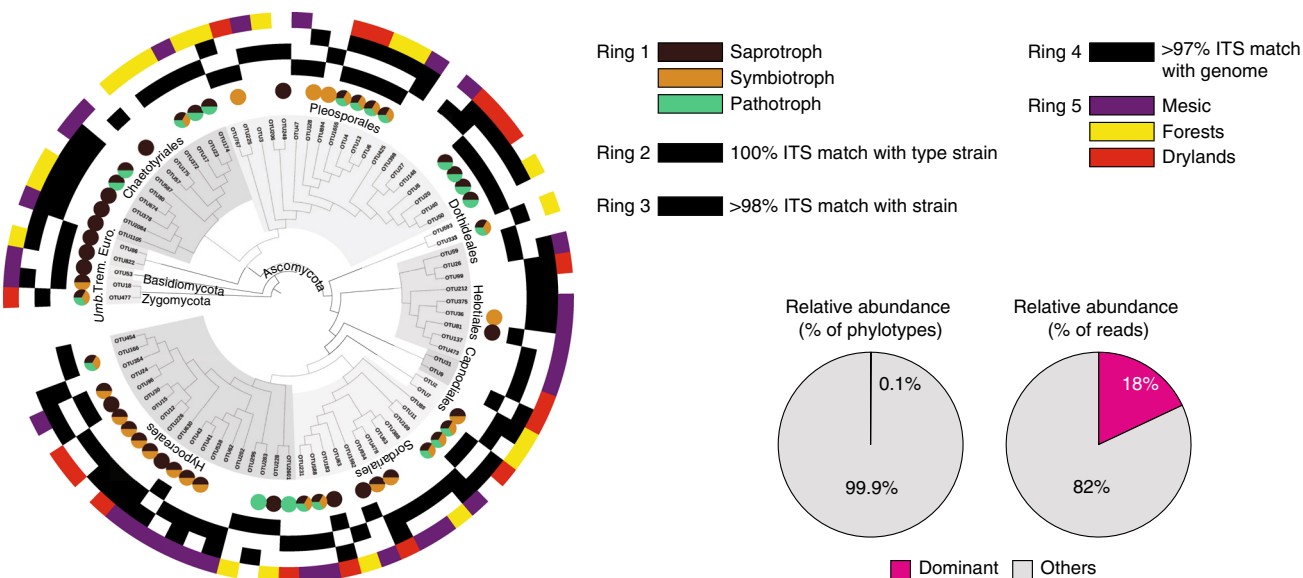

**Fig. 1** Phylogenetic distribution of the 83 dominant fungal phylotypes. Ring 1 indicates the most likely functional guild for each phylotype, inferred by parsing the phylotypes with FUNguild[78], where brown = saprotroph, yellow = symbiotroph, green = pathotroph. For each phylotype, black squares in Ring 2 and 3 indicate whether there is a representative type strain or isolate match at ≥98% ITS2 gene sequence similarity level, while black squares in Ring 4 highlights the presence of a matching genome at ≥97% similarity threshold. The colouring on the outermost ring matches the preferential environmental distribution for all phylotypes (n = 83), where purple = mesic, yellow = forest, red = drylands. Trem. = Tremellomycetes, Umb. = *Umbelopsis*, Euro. = Eurotiomycetes. The abundance (%) and richness of the dominant phylotypes in relationship to the non-dominant phylotypes are reported in the pie charts. The nucleotide sequence of the dominant phylotypes is reported in Supplementary Data 1

validated the results of the dominant phylotype identification using data from a recent biogeography study[14] that surveyed the fungal community from soils worldwide (Supplementary Note 1).

The variation in Bray-Curtis dissimilarity index of the dominant fungal community was positively correlated with the diversity of the non-dominant fungal community ($R^2 = 0.53$, $p < 0.001$, Supplementary Fig. 3). Individual phylotype abundance and frequency were positively correlated (Pearson $r = 0.50$, $p < 0.001$; Supplementary Fig. 4A), with relatively few phylotypes being both highly frequent and abundant, and the majority of the retrieved fungal taxa restricted in their abundance (Supplementary Fig. 4B).

When compared against the GenBank database, 33 out of the 83 dominant phylotypes had 100% sequence identity with sequences obtained from fungal type material, while 47 phylotypes had ≥98% match with sequences from fungal cultures. Two phylotypes matched uncultured organisms. The most ubiquitous fungi found in our dataset were members of the Pezizomycotina, namely Sordariomycetes (including members of the genera *Podospora*, *Chaetomium*, *Fusarium*, and *Trichoderma*), Leotiomycetes (genera *Leohumicola*, *Talaromyces*, *Cadophora*) Eurotiomycetes (genera *Penicillium*, *Knufia* and *Exophiala*) and Dothideomycetes (genera *Alternaria*, *Aureobasidium*,

*Cladosporium*, *Curvularia*), together with two members of the Tremellomycetes (genera *Vishniacozyma* and *Saitozyma*) and one Mucoromycotina (genus *Umbelopsis*) (Supplementary Data 1). Overall, Ascomycota was also the most phylotype-rich and abundant lineage in our dataset, followed by Basidiomycota (~15%) and Unclassified Fungi (d:Fungi, ~20%) (Supplementary Fig. 5A-B), which rather tended to be locally abundant but relatively infrequent (Supplementary Fig. 6A-C).

**Ecological drivers of dominant fungal phylotype distribution.** On average, the 83 dominant fungal phylotypes showed similar levels of relative abundance among the sampled habitats, representing 15–27% and 10–25% of the total number of reads in different biomes and continents, respectively; the highest average abundance was found in tropical and temperate forests (Fig. 2a). Subsequently, we sought to identify the major predictors of the distribution of dominant fungal taxa worldwide. We used ecological networks and semi-partial correlation analyses to group dominant taxa by their environmental preferences, and found three dominant ecological clusters, grouping ~70% of all fungal phylotypes, that were associated to three different biomes: mesic (higher Aridity Index, meaning higher precipitation), forests

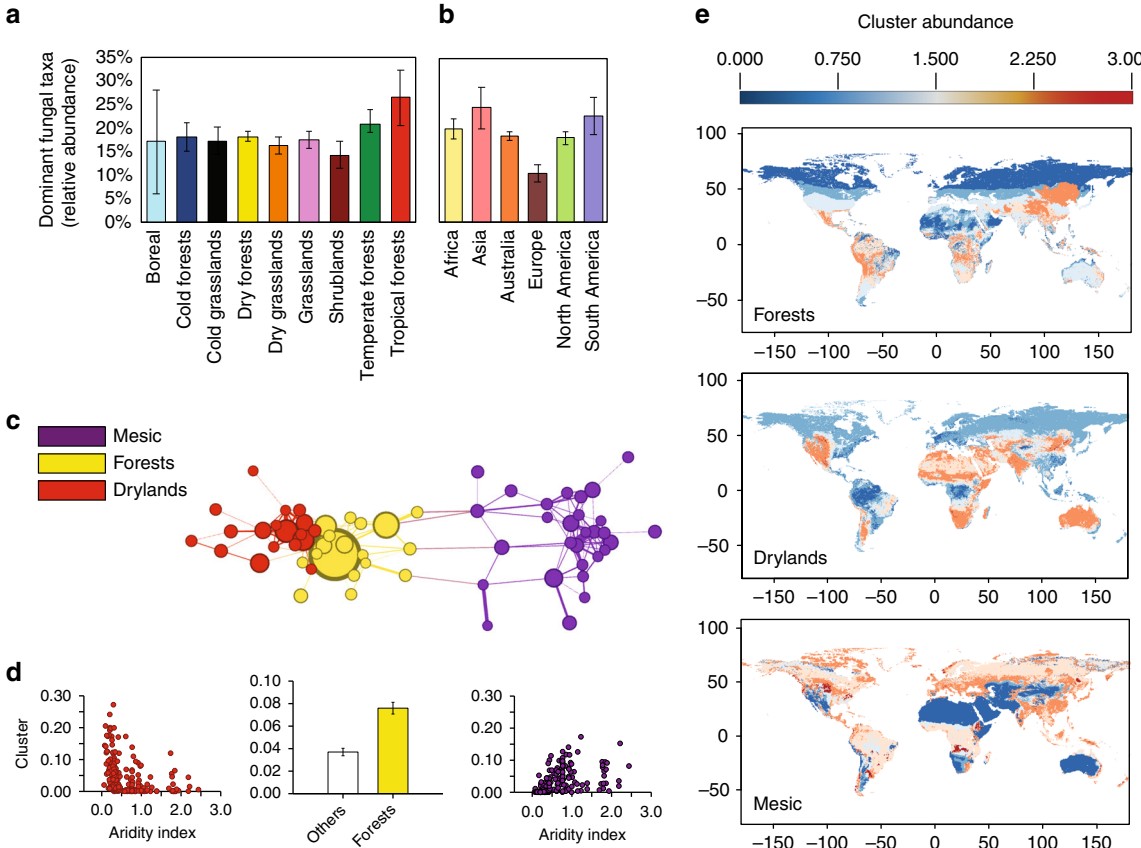

**Fig. 2** Distribution and habitat preferences for dominant fungal phylotypes. Relative abundance (mean ± SE) of dominant phylotypes across (**a**) boreal ($n = 3$), cold forests ($n = 18$), cold grasslands ($n = 22$), dry grasslands ($n = 42$), dry forests ($n = 60$), grasslands ($n = 41$), shrublands ($n = 15$), temperate forests ($n = 27$), and tropical forests ($n = 7$), and (**b**) continents. Biome classification followed the Köppen climate classification and the major vegetation types found in our database. Grasslands include both tropical and temperate grasslands. Shrublands include polar, temperate, and tropical shrublands. **c** Network diagram with nodes (fungal phylotypes) coloured by each of the three major ecological clusters (i.e., mesic in purple, forests in yellow, drylands in red) that were identified, highlighting that phylotypes within each ecological cluster tend to co-occur more than expected by chance. **d** Relationships between the relative abundance of the phylotypes assigned to each ecological cluster and their major environmental predictors. **e** Predicted global distribution of the relative abundances of the three major ecological clusters of fungal phylotypes sharing habitat preferences for drylands, forests and mesic ecosystems. We found a positive (Spearman) correlation between predicted and observed data for drylands ($r = 0.47$; $p < 0.001$), forests ($r = 0.32$; $p < 0.001$) and mesic ($r = 0.61$; $p < 0.001$) ecosystems. The scale bar represents the standardised abundance (z-score) of each ecological cluster

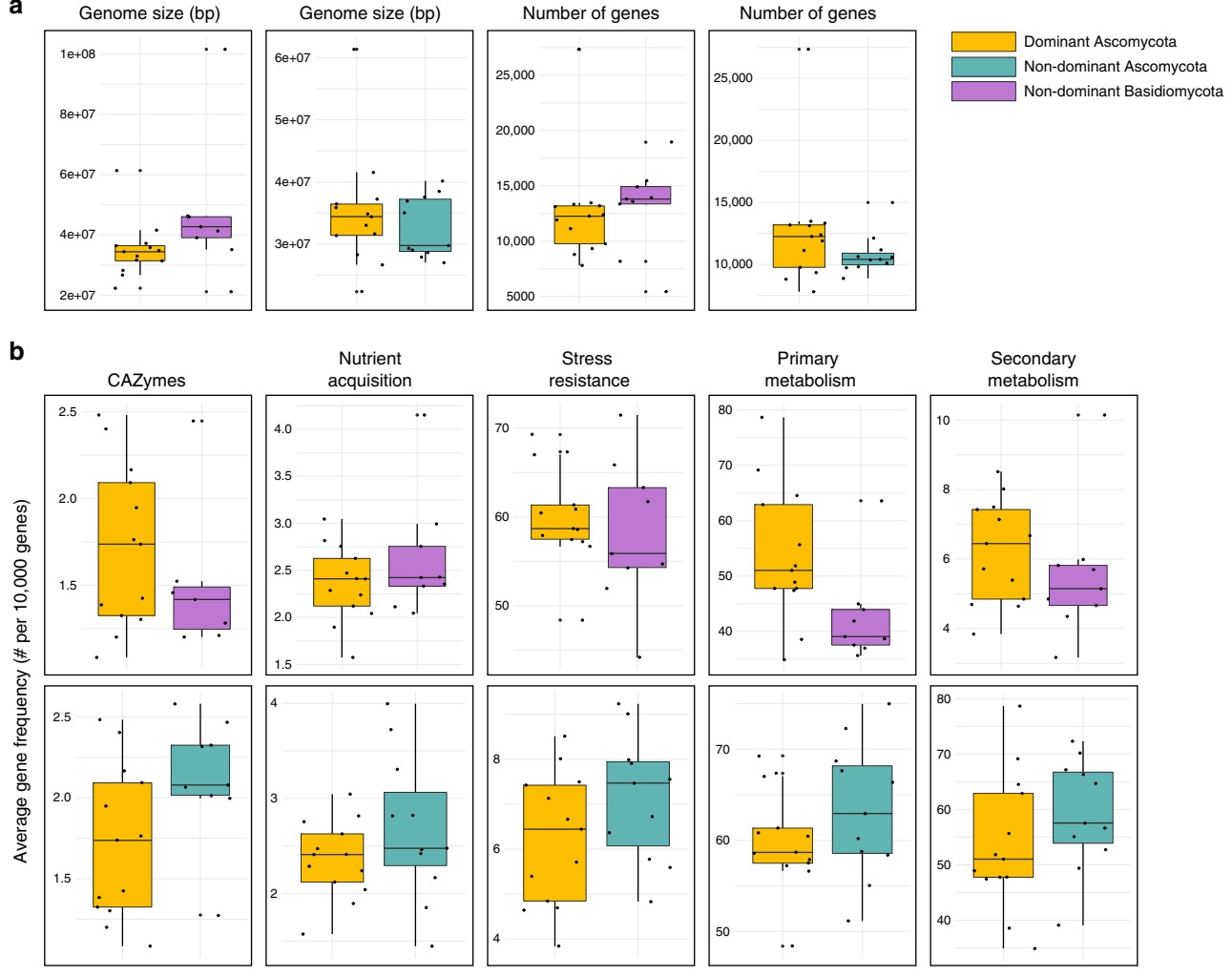

**Fig. 3** Differences in abundance and occurrence of the functional genes across phyla. **a** Boxplots showing the average genome sizes and number of putative genes of the dominant Ascomycota ($n = 13$) compared with non-dominant Basidiomycota ($n = 9$), and dominant Ascomycota compared with non-dominant Ascomycota ($n = 11$) representatives. **b** Boxplots showing the average relative frequency of gene classes associated with different ecological traits in dominant Ascomycota vs. non-dominant Basidiomycota (upper row), and dominant Ascomycota vs. non-dominant Ascomycota (lower row). Source data are provided as a Source Data file

(both mesic and dry forests) and dryland (low Aridity Index) (Fig. 2c, d; extended results of the mapping of ecological clusters are reported in Supplementary Table 1). Significant associations (e.g., Aridity Index: Mesic: $r = -0.47$; $P < 0.001$; Dry: $r = 0.39$; $P < 0.001$) were also obtained when controlling for spatial auto-correlation (i.e., using latitude and longitude as controlling matrix). Consistently, the fungal maps estimating the expected geographical distribution and abundance of dominant soil fungal phylotypes (Fig. 2e), broadly reflected the extent of well-characterised mesic, dry and forest biomes. We further corroborated our predictive maps of the dominant soil fungi distribution using continental-scale datasets for soil microbial distribution. Results from these cross-validations are detailed in Supplementary Note 2.

**Whole-genome comparisons.** Further to the above analyses, we investigated whether functional gene information can explain why Ascomycota phylotypes are more dominant that Basidiomycota phylotypes, and also why some Ascomycota phylotypes are more dominant than others. Using Random Forest analyses, we identified functional genes from available sequenced whole genomes that characterise (1) dominant Ascomycota taxa vs.

other non-dominant Ascomycota taxa and (2) dominant Ascomycota taxa vs. other non-dominant fungal taxa from the phylum Basidiomycota. We found a number of ecologically relevant genomic traits that varied significantly between the dominant and non-dominant members of the soil fungal community (Fig. 3). Despite the larger genome size and coding gene number in the Basidiomycota (Fig. 3), after standardization of gene content to genome size a significantly higher number of genes associated with nutrition (e.g., phosphate transporter, nitrogen immobili-sation), and carbohydrate metabolism (e.g., CAZymes related to degradation of complex sugars, polysaccharide synthesis, and catalytic efficiency enhancing) characterised the dominant Ascomycota (Supplementary Fig. 7). Additionally, when compared with Basidiomycota, the Ascomycota representatives in our dataset exhibited significant higher frequency of genomic traits associated with both stress-tolerance and competitive abilities, such as melanin deposition, as well as resistance to antibiotics and antibiotic production. However, such stark genomic contrast was not significant when the genomes of dominant Ascomycota and non-dominant Ascomycota were compared, with the majority of the analysed traits being non-significantly different in terms of standardised abundance (Fig. 3, Supplementary Fig. 8).

## Discussion

Understanding why some species display larger geographic distribution than others is a major goal in ecology. Disentangling patterns of dominance can help to elucidate the relationships between organisms and their environment, as well as the forces shaping biodiversity and co-existence dynamics[22]. Importantly, efforts to catalogue dominant species among animal and plant communities have supported the development of fundamental unifying ecological principles[23], that are the ultimate foundation of prioritisation strategies for conservation and management of biodiversity on a global scale[24]. Yet, equivalent efforts are rarely applied to microbial communities[25]. Identifying such patterns of dominance is particularly important for soil-inhabiting fungi, one of the ecologically most relevant groups of living organisms. In this study, we addressed this knowledge gap by characterising the identity, global distribution and ecology of dominant fungi identified in soil samples collected from across the world (Supplementary Fig. 1).

Surveying soil fungi across wide spatial scales poses a series of methodological and technical challenges. The accurate estimation of a taxon occurrence on a global scale requires to analyse a wide array of biomes, climates, and continents across the globe. Our sampling campaign covered the nine most common terrestrial biomes of the world, surveyed across 235 sites, from 18 countries and 6 continents, making the present global database one of the most inclusive for fungi (cfr.[14]). Additionally, as fungal communities are vastly diverse, adequate sampling depth is a critical requirement to obtain an accurate evaluation of fungal community composition. In this study, we were able to obtain a total of 12M reads, and an average of 47,207 reads/565 phylotypes per sample, resulting in mostly saturated species rarefaction and accumulation curves (Supplementary Fig. 2A, B), which is indicative of a satisfactory representation of the most common members these communities harbour. Thus, although we are certainly underestimating the actual diversity of soil-inhabiting fungi, we argue that the comprehensive sampling effort and methodology used here are adequate to explore the most common members of the soil mycobiome.

To detect and characterise the globally dominant soil taxa, we selected those fungal phylotypes that were abundant (top 10% most common phylotypes sorted by their percentage of the total ITS rRNA reads), frequent (i.e., occurring in at least one third of the samples from a given biome), and had the highest habitat breadth (i.e., being dominant in at least half of the sampled biomes). We found that less than one hundred (<0.1% of the retrieved fungi) fungal phylotypes are dominant in soils across the globe. Unsurprisingly, phylotype abundance and frequency were positively correlated, with relatively few phylotypes being both highly frequent and abundant, and the majority of the retrieved fungal taxa restricted in their distribution. Similar correspondence between abundance and geographical distribution have been recently observed for dominant bacteria in soil[25], suggesting that both soil prokaryotic and eukaryotic microbial communities broadly parallel patterns of dominance previously documented in plant and animal communities[26,27], with locally abundant taxa tending to occur in a greater proportion of sites and to have wider geographic distributions.

Most of the identified dominant fungal phylotypes belonged to a single phylum: Ascomycota. Previous reports identified the global abundance and distribution of ascomycetes in soil systems globally[14,16,28]. To the best of our knowledge, this is the first study to show the extensive dominance of a few dozen phylotypes from fungal communities across a broad range of soils and environmental conditions globally. However, the diversity of the dominant taxa identified in this study is limited to the fungal taxa amplified with the primer pair used here, and to the biomes included in our survey. For example, we are potentially underestimating the distribution of some fungal lineages that are poorly amplified with commonly used ITS primers, such as members of Glomeromycota and Archaeorhizomycetes[29]. Importantly, our dataset had limited representation of boreal and tropical systems, possibly limiting the number of dominant fungal taxa characteristics of these biomes. Therefore, we envisage that future studies including under-sampled regions of the world will allow to identify more of the common members of the soil mycobiome.

Apart from two phylotypes associated with the genera of yeast fungi *Vishniacozyma* and *Saitozyma*, respectively, most of the dominant taxa retrieved here belonged to culturable genera of free-living filamentous fungi, suggesting that functional traits associated with this group may play an important role in defining dominance relationships[30]. Our results are supported by earlier studies that identified many of these genera amongst the frequently recorded members of the soil mycobiome[5,31], with the majority of the most ubiquitous and abundant fungi at the global level being from relatively well characterised fungal lineages. By contrast, the unclassified fungi (d:Fungi), although comprising ~20% of the retrieved phylotypes (Supplementary Fig. 5A), overall exhibited a narrow distribution (Supplementary Fig. 6C), suggesting that these poorly characterised phylotypes are not dominant on a global scale, and represent locally abundant, but relatively infrequent members of the soil biosphere.

The dominant fungi identified in this study showed clear environmental preferences, and were associated to three different biomes: mesic, forests, and drylands. Interestingly, unlike dominant bacteria[25], soil properties (e.g., soil pH) were poor predictors of the relative abundance of dominant fungal taxa. This result is consistent with other global studies, which reported a significant correlation of pH and soil elements with particular fungal functional groups (e.g., mycorrhizal fungi[14,32–34]) and a generally minor influence of edaphic characteristics on other fungi[14,16]. In fact, climate is often reported as the most important environmental factor predicting fungal community composition[14,16]. Such differences in habitat preferences for dominant fungi confirm the importance of vegetation and climatic parameters (i.e., aridity index, precipitation/evapotranspiration) in determining the composition and community assembly dynamics of fungi in soil[14,16,35].

The co-occurrence patterns of the dominant taxa showed no obvious association between coarse-level (i.e., class to family) taxonomic identity and habitat preferences, with most of the orders/families being represented in each main ecological cluster. However, at the genus level, the dominant phylotypes exhibited clear differences in ecosystem preferences, mainly ascribed to differences in climatic conditions. Such result supports the idea that, at least at a global scale, fungal communities might be more vulnerable to climatic changes than bacterial communities, which are often associated to soil properties[36]. This implies that climatic changes and increasing landscape fragmentation could result in changes to dominance hierarchies in fungal communities[37]. Our results also suggest that the relative contribution of dominant soil fungi to ecosystem services may change in contrasting regions of the globe, as indicated by their preferential distribution. These findings provide interesting opportunities for testing hypotheses on structure–function relationships in natural soil communities. For example, they could be used to set up manipulative experiments (such as coalescence experiments, or enrichment/removal manipulations reproducing particular sequences of species addition/loss) targeting soils wherein these taxa vary in their abundance in different regions. These experiments would significantly increase our understanding of the relative importance and ecological role of this prominent group of soil-dwelling fungi in contrasting environments, allowing for better predictions on climate-mediated biodiversity and functionality shifts.

Mechanistically, several reasons could explain the dominance of few ascomycetes compared to other fungi across the globe, including dispersal abilities, life-styles and functional attributes. First, global dominance implies the capacity of a taxon to disperse trans-continentally, with passive wind dispersal likely being the most efficient mechanism for such long-distance spreading[38]. Our data indicate that globally distributed fungal phylotypes include well known wind-dispersed fungal genera, such as *Alternaria, Aureobasidium, Cladosporium, Penicillium, Fusarium, Chaetomium, Acremonium*, and *Curvularia* (Fig. 1). These taxa are often detected in high abundance in airborne samples from different origins, including organic and inorganic dust, and atmospheric aerosol particles[12,39–42]. However, wind dispersal abilities are not exclusive of the dominant taxa retrieved in this dataset. For example, many members of Basidiomycota are capable of long-distance dispersal[43], are often recovered in high abundance in aerial samples[41,44], and are considered 'core' members of the airborne biological matter[45]. Yet Basidiomycota, the second phylum in terms of abundance and richness in our dataset (Supplementary Fig. 3A-B), comprised only a small fraction of the dominant phylotypes (<2%, Fig. 1), with many basidiomycetous phylotypes being dominant locally, rather than at the global scale (Supplementary Fig. 5A-B). Moreover, none of the globally dominant Basidiomycota belonged to soil-inhabiting Agaricomycetes, whose members are also often retrieved in airborne samples[41,44,46–49], and could contribute to about 50% of all fungal spores in the atmosphere[12,39,40]. Therefore, we argue that wind dispersal ability alone is insufficient to explain the almost exclusive cosmopolitan distribution of few ascomycetes globally. Conversely, other factors, such as life-styles and ability to colonise multiple niches, can also play important roles.

We found that many of the dominant phylotypes were associated with taxa of ecological, agricultural, and medical importance, and were characterised by multiple trophic modes. Notably, in addition to their saprobic life-styles (Fig. 1), members of the genera *Fusarium, Alternaria* and *Chaetomium* comprise several species of opportunistic potential plant pathogens affecting agriculture and horticulture globally[6,50], and have increasingly been found to also affect humans[51]. Similarly, the eurotiomycetous genera *Cladophialophora, Knufia* and *Exophiala* harbour saprotrophic species on plant debris and clinically relevant agents, and occur in both natural and anthropogenic habitats[52], while *Metarhizium* spp. are insect-pathogenic fungi for a wide range of hosts, and can also colonise plant roots[53]. Given the remarkable versatility of interactions exhibited by these fungal lineages, we hypothesize that possessing flexible trophic capabilities may allow some dominant taxa to occupy multiple environmental niches, and may be an important additional factor in defining fungal habitat breadth, and thus dominance, in soils.

The majority of these dominant fungi were characterised by higher genomic potential for resource utilisation, competition and stress tolerance compared with other ecologically important fungi, such as saprobic Basidiomycota. While our analyses were based on a few available genomes and on sequenced representatives of each species rather than the whole of the phylotypes identified, we still observed significant trait differences between these dominant and non-dominant phyla: the Ascomycota representatives in our dataset exhibited significant higher frequency of genomic traits associated with both stress-tolerance and competitive abilities, such as melanin deposition, as well as resistance to antibiotics and antibiotic production. Interestingly, antibiotic production has been identified as a global determinant of inter-kingdom (i.e., fungi-bacteria) biotic interactions in topsoil[4]. Our results hint at the possibility that such trait might confer competitive advantages also at the intra-kingdom level, but further studies are needed to corroborate this hypothesis. Overall,

the marked variation in genomic potential between dominant Ascomycota and non-dominant Basidiomycota (Fig. 3, Supplementary Fig. 5) suggests that ascomycetes may be better equipped to withstand environmental stresses and are able to utilise a higher number of resources, leading to more generalist strategies that may contribute to their increased dominance in soils. It will be of interest, as further genomes for dominant and non-dominant soil fungi become available, and genomes of multiple isolates within each phylotype are produced, to understand if these genomic patterns still hold true.

Interestingly, we did not observe the same stark genomic contrasts among the genomes of dominant Ascomycota and non-dominant Ascomycota. As our comparison was mainly restricted to saprobic fungi, the lack of strong changes in the relative number of genes could possibly indicate that these genomic traits are highly conserved at the intra-phylum level for fungi with similar life-styles[54]. It is also plausible that gene regulation and expression mechanisms, rather than number of genes per se, contribute to determine process rate, and thus fitness and adaptability[55,56]. However, the limited number of published genomes and the scarce information on other genes or class of genes not considered in this study hamper our ability to comprehensively assess the role of functional differences in explaining observed within-phylum dominance patterns. Further analyses including more genomes from the dominant phylotypes will allow us to reveal possible attributes underpinning the wide geographic distribution of the suite of dominant fungi found in our study. Our analyses here provide information on which of these genomes should be included in future sequencing efforts. As several of these dominant soil phylotypes are from culturable taxa of ecological, agricultural, and medical importance, their genome sequencing will not only improve our understanding of which genomic traits are associated with dominance within an ecosystem, but also provide new tools to a wide range of scientific disciplines.

Finally, given the strong relationship between soil fungal communities and ecosystem functions[57], and the correlation between dominant species and β-diversity of whole fungal communities, we posit that shifts in dominant fungal taxa should be taken into account when predicting changes in ecosystem functions under climate change and increasing habitat fragmentation. Taken together, our findings provide a baseline understanding of dominant fungal identity, distribution, and ecological attributes in global soils. This understanding is critical if we are to develop approaches and strategies aimed at preserving soil microbial diversity and functionality worldwide.

## Methods

**Identification of dominant phylotypes**. Details of sample locations and collection have been reported and described previously[25]. In brief, bulk soils were separated from plant roots from 235 soil samples collected across 18 countries, covering nine biomes (temperate, tropical and dry forests, cold, temperate, tropical and arid grasslands, shrubland, boreal) across the globe (Fig. S1). The extracted DNA samples were frozen and shipped to the Next Generation Genome Sequencing Facility of the University of Western Sydney (Australia). Fungal diversity was determined by sequencing the Internal Transcribed Spacer (ITS) region 2[58] with primers FITS7 (GTGARTCATCGAATCTTTG)/ITS4 (TCCTCCGCTTATTGA-TATGC)[59] on a Illumina MiSeq platform (2 × 300 PE), and both positive and negative controls were included. All reads were quality filtered and dereplicated with the USEARCH pipeline, and low-quality bases were end-trimmed before merging. All the merged reads had an expected error < 0.5; the quality-filtered reads were clustered into operational taxonomic units (OTUs) or phylotypes of the length of 180 bp, at both the 97 and 100% similarity thresholds using UPARSE[60] and UNOISE[61], respectively. Phylotype identification was obtained against the UNITE fungal database (V7.2)[62] using the SINTAX algorithm with a ≥80% probability threshold[63].

More than 99% of the total 22,209 and 24,137 phylotypes obtained with the 97% and the 100% clustering methods, respectively (~12 M reads), were identified as belonging to the kingdom Fungi at confidence level ≥ 80% with the SINTAX algorithm. The unclassified phylotypes either belonged to lineages that are poorly

characterised and represented in the UNITE database (d: unclassified), or belonged to other eukaryotic members of the soil community (d: Protista, and d: Plantae). The two clustering approaches (97 and 100% similarity threshold) produced considerably overlapping communities, with alpha and beta diversity metrics being highly positively correlated (Spearman's $r > 0.88$ for observed richness, and $r > 0.95$ for Bray-Curtis dissimilarity value) (Supplementary Fig. 9), and the 100% similarity threshold only relatively increased the resolution of fungal diversity (1.2 times OTU richness), in line with recent systematic investigations that demonstrated the limited effect of the OTU clustering cut-off on the resulting fungal community structure[64].

Although the 97% similarity threshold is widely accepted to define molecular OTUs from the fungal ITS region[65], the use of the amplicon sequence variants has the important advantage to allow for the direct comparison of sequences across studies[66]. Therefore, given the substantial overlap between the communities, we decided to use the OTUs clustered into phylotypes at the 100% similarity threshold for all the downstream analyses. To detect and characterise the globally dominant soil fungal phylotypes, we selected those fungal OTUs that were abundant (top 10% most common phylotypes sorted by their percentage of the total ITS rRNA reads), frequent (i.e., occurring in at least one third of the samples from a given biome), and had the highest habitat breadth (i.e., being dominant in at least half of the sampled biomes). Additionally, to confirm that the OTU clustering cut-off did not bias the obtained results, the analysis was repeated on the OTUs clustered at the 97% threshold (the obtained dominant phylotypes are reported in Supplementary Table 2). On average, a single phylotype was observed in five different samples (mean), with ~72% of the phylotypes being found in ≤2% of the samples, and only 33 OTUs being found in ≥50% of the samples (Fig. S2A). Each phylotype sequence from the dominant community ($n = 83$) was searched against the GenBank repository using the BLAST function, and a representative sequence was selected for each OTU using a 99% similarity cut-off. An additional search against the UNITE online database (https://unite.ut.ee/) was performed and the closest Species Hypothesis (SH) for each phylotype was recorded (Supplementary Data 1). The representative sequences were aligned and a maximum likelihood phylogenetic tree was built with the MEGA7 software[67] using a GTR substitution model of nucleotide evolution. The tree was then visualised and annotated using iTOL V3[68].

**Identification of shared habitat preferences**. To identify ecological clusters (modules) of strongly associated dominant fungal phylotypes, a correlation network (i.e., co-occurrence network) was established. Our network includes the reported 83 dominant fungal taxa. We then calculated all pairwise Spearman's rank correlations (ρ) between all soil fungal taxa. We focused exclusively on positive correlations because they provide information on fungal taxa with similar environmental preferences[69,70]. We considered a co-occurrence to be robust if the Spearman's correlation coefficient was $>0.40$ and $p < 0.01$. The network was visualised with the interactive platform gephi[71]. Finally, we used default parameters from gephi to identify ecological clusters (modules) of fungal taxa strongly interacting with each other. We then computed the relative abundance of each module by averaging the standardised relative abundances (z-score) of the taxa that belong to each ecological cluster.

We also aimed to identify the environmental preferences of the different ecological clusters. To do this, we conducted semi-partial correlations (Spearman) using the ppcor package[72] between the relative abundance of ecological clusters and 14 environmental predictors: climate variables (Aridity Index, minimum and maximum temperature, precipitation seasonality and mean diurnal temperature range—MDR), UV radiation, net primary productivity (NDVI index, 2003–2015 period), soil properties (texture [% of clay + silt], soil pH, total C, N and P concentrations and C: N ratio) and dominant ecosystem types in our dataset (forest and grasslands). Ecosystem types were coded as categorical variables with two levels: 1 (a given ecosystem type) and 0 (remaining ecosystem types). In order to exclude possible confounding effects due to spatial autocorrelation of environmental variables, we additionally repeated the correlation analysis between the main ecological factors and phylotype relative abundance, while controlling for space (latitude and longitude). Methodological details related to these environmental data are given in[25].

**Identification of genomic traits across phyla**. Of the dominant 83 phylotypes, 43 had ITS2 sequences that BLASTed with >97% identity to a type specimen with a sequenced genome. Accounting for duplicated species, these 43 phylotypes accounted for 24 separate species. Of these, 13 had publically available genomes (see Supplementary Data 2). Similarly, 11 and nine non-dominant Ascomycota and Basidiomycota phylotypes, respectively, had publically available genomes with ITS2 sequences that had >97% identity with our recovered OTUs (Supplementary Data 2). From these genomes, we downloaded gene predictions for CAZymes, peptidases, secondary metabolism as well as general genes annotated to GO, KEGG, and KOG categories from the Mycocosm database curated by the Joint Genomes Institute[73] (last accessed August 2018). We also identified all genes within each genome associated with stress tolerance and nutrient acquisition that have been previously identified as fungal traits driving ecosystem dynamics[30]. Annotation of antibiotic resistance genes and resistance mechanisms was performed using the Resistance Gene Identifier (RGI) algorithm and the Comprehensive Antibiotic Resistance Database (CARD)[74]. Briefly, the gene coding

sequence of all 'Best Filtered Models' for each genome were downloaded from Mycocosm and run on the RGI pipeline using 'Perfect, Strict and Loose' hits and 'High Quality/Coverage' sequencing quality. Genomes were annotated for resistance class, drug class, antibiotic resistance gene class and best hit antibiotic resistance ontology.

Finally, we used Random Forest analyses to identify particular functional gene from available genomes characterising (1) dominant Ascomycota taxa vs. other non-dominant Ascomycota taxa and (2) dominant Ascomycota taxa vs. other non-dominant fungal taxa (Basidiomycota). In these analyses, functional genes were used as predictor variables and (1) dominant Ascomycota/other non-dominant Ascomycota or (2) dominant Ascomycota taxa/other non-dominant fungal taxa (Basidiomycota) as response variable[75]. These analyses were conducted using the rfPermute package[76] in R (https://www.r-project.org). Differences in abundance and occurrence of the functional genes identified with the Random Forest technique among the different groups (i.e., dominant Ascomycota vs. non-dominant Basidiomycota, and dominant Ascomycota vs. non-dominant Ascomycota) were investigated using one-way anova, as implemented in the R package mvabund[77].

**Reporting summary**. Further information on research design is available in the Nature Research Reporting Summary linked to this article.

## Data availability
The raw reads are available at https://figshare.com/s/9772d31625426d907782 (https://doi.org/10.6084/m9.figshare.5923876). All other relevant data is available upon request.

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

## Acknowledgements

We would like to thank Noah Fierer and Alberto Benavent-González for providing some soil samples, Victoria Ochoa and Beatriz Gozalo for their help with soil analyses, and Andrew Bissett for providing access to the ITS2 data of the Australian Microbiome Project. E.E. and B.K.S. were supported by the CRC-CARE project 4.2.06–16/17; B.K.S. was also supported by the Australian Research Council (DP 170104634 and DP190103714). M.D.-B. acknowledges support from the Marie Sklodowska-Curie

Actions of the Horizon 2020 Framework Programme H2020-MSCA-IF-2016 under REA grant agreement no. 702057; J.P. would like to acknowledge the Australian Research Council for research funding (DE150100408). The work of F.T.M. and the global drylands database were supported by the European Research Council (ERC Grant Agreements 242658 [BIOCOM] and 647038 [BIODESERT]) and by the Spanish Ministry of Economy and Competitiveness (BIOMOD project, ref. CGL2013–44661-R). R.D.B. was supported by the UK Department of Environment, Food and Rural Affairs (DEFRA) project number BD5003 and a BBSRC International Exchange Grant (BB/L026406/1)

## Author contributions

M.D.-B. and B.K.S. conceived the study, M.D.-B., F.T.M., D.J.E., R.D.B. and B.K.S. collected the soil samples, E.E., M.D.-B., J.M.P. and J.W. performed the data analysis. E.E., M.D.-B. and B.K.S. wrote the manuscript, with input from all the co-authors.

## Additional information

**Competing interests:** The authors declare no competing interests.

