## [Peer Review File · Nature Communications]

Reviewers' comments:

Reviewer #1 (Remarks to the Author):

This manuscript present information on the distribution and ecological preference of fungi at a global scale, analysing 235 soil samples collected across the globe.

This is an interesting and appealing manuscript. While I do like the approach and the results presented, I do have a number of comments related to interpretation, analysis and presentation of the results. In particular, the identification of the dominant soil fungi needs further explanation (e.g. why these 83 fungal taxa were selected and not the 33 taxa found in > 50% of the samples (see below for specific comments). This because interpretation of the results possibly depends very much on which taxa are included.

Specific comments:

Please provide additional information on the rationale as to why 83 dominant taxa were selected (e.g. why not 95 taxa). It is important to explain this in detail because the conclusions of this paper depend on the number (and identity) of taxa selection. In this sense, it would help to include a rank-abundance diagram for the 150 (or 200) most abundant taxa (Figure S2b is not so clear because all taxa are included). This shows the reader why these 83 taxa were included (perhaps the 83 taxa identified could also be coloured red in Figure S2a).

Would it be possible to present a phylogenetic tree (including genetic distance) of the 83 dominant taxa in the Supplement (I wonder how many taxa are genetically highly similar/ genotypes of the same species (e.g. very high sequence similarity).

Line 129-142 presents a discussion on the identity of the dominant fungi. The primer choice has a large impact on fungal community composition (e.g. some "dominant" fungi may not be detected with the ITS2 primer used). Please add a qualifier to inform the reader about this issue.

Line 140 states that mainly filamentous fungi were detected. What about yeasts?

As far as I know Results and Discussion are separate sections for Nature Communications.

Line 166 refers to three global survey of soil fungi (ref 8, 20, 21). Probably it is good to explicitly state what makes this study different from those three studies (e.g. this is nicely explained in the Supplement on page 13 and possibly there are other arguments as well).

Line 170 states that soil properties (e.g. soil pH) were poor predictors of the relative abundance of dominant fungal taxa. Is there also no effect of soil pH (and soil properties) on fungal community composition? Several studies do indicate that soil pH is a major driver of fungal community composition. Any explanation for the absence of any clear effect?

Line 211: phylotype richness was highest in the Ascomycota: I would be good to formally show that the Ascomycota still are the most abundant phylum if there is a correction for the number of Ascomycota fungal taxa? Also is there any evidence that the primer used has higher specificity for the Ascomycota (if not, it would be good to state that)

Line 265: resistance to antibiotics: Bahran et al. 2018 (Nature) also found effects of antibiotics if I am right. Please verify and discuss this.

Line 319-325: The future outlook to use manipulative experiments is an interesting and relevant one. What is the role of dominant soil fungi in structuring fungal community composition and fungi-fungi interactions? Perhaps this can be discussed as well. It would also be interesting to know whether some of the dominant soil fungi can be considered as keystone taxa, or at least discuss this issue (see review by Banerjee et al. 2018, Nature Reviews Microbiology about keystones).

Line 346: "the and the"???

Line 345-350: Please provide further information on the data quality and analysis (e.g. sequencing depth per sample; rarefaction curves per sample)

Line 373: "33 OTUs being found in > 50% of the samples": Are the conclusions of this study different if those 33 OTUs are taken instead of the reported 83 OTUs.

Line 385: "co-occurrence networks": Is it worthwhile to search for keystone taxa (OTUs)? This might reveal some further interesting patterns.

Line 468: year of the reference is missing.

Please provide a map with sample locations (in the supplement)

Reviewer #2 (Remarks to the Author):

Egidi et al. report their analysis on the global spatial distribution of the dominant soil fungi using ITS rRNA genes. The analysis and questions in this study are very similar to what the authors published recently on bacteria (Science 359: 320-325, 2018). Similarly, to their previous paper, the authors focused on a number of dominant fungal taxa found in their soil samples.

Although I appreciate the usefulness of such large-scale surveys, I believe that the value of this study resides mostly in the dataset generated rather than the reported patterns, as most findings have been reported previously both on local and global scales. The authors have attempted to highlight the novel aspects of their study in several places in the manuscript, but to me the novelty is limited to the data analysis and presentation and not the findings. For example, network analysis reveals the co-abundance behavior of fungal taxa in different habitats, and correlations simply reflect variation in the abundance of certain soil fungi across different habitats. Such findings are not novel, as any comparison of distinct microbial habitats would reveal such trends.

I have also concerns about the generalization of the findings reported by the authors, most importantly due to geographic limits in samples and a strong spatial dependence between samples. In spite of sampling 235 sites, vast global areas were not accounted for (e.g., tropical Africa and Asia) and most sampled sites are spatially aggregated, i.e. covering the same plant community (mostly steppe or shrubland). These samples are thus highly spatially autocorrelated. As these are unlikely to be independent, conclusions derived from them are thus prone to a type I error. The multicollinearity of space and environment could have been minimized by a more appropriate sampling design.

In addition, the authors found only 10 dominant OTUs shared between the two datasets, which shows that dominant taxa uncovered in such datasets depend strongly on methodology or sampling coverage. Was there any correlation between the abundance of the shared dominant OTUs in the two datasets?

Please also report the % identity of the shared OTUs. The results of the cross-validation should be reported in the main text.

Although potentially interesting, the results based on the genomic data are mostly unreliable. First many of the dominant taxa studied are not included in this analysis and the conclusions are driven based on a few available genomes, which make many of the conclusions biased. For example, the abundant taxa that are easily cultured have representative genomes, and they are most likely to be generalists as most genomic data come from non-soil fungi. In addition, even 3% dissimilarity in rRNA may indicate dissimilar genomic content, as fungal genomes show interspecies variability. Such analysis also oversimplifies the geographic trends in genomic potential and expression (e.g. Ellison, Christopher E., et al. *Proceedings of the National Academy of Sciences* (2011): 201014971; Branco, Sara, et al. *Molecular ecology* 26.7 (2017): 2063-2076.). And could the authors explain why they only compared Ascomycota and Basidiomycota? What about other fungal phyla? For example, Zygomycota which seem to be rare in this study.

L84-L85 This overlooks a number of efforts that have addressed the biogeography of fungi on large scales.

L85- L86 This sentence is vague. How many, what proportion, at what taxonomic levels?

L93- L94 Unclear how the authors reached this.

L97- L100 All these would apply to studying communities as well. Do the authors believe that studying dominant taxa would reveal different results compared to community studies? I believe not.

L101- L103 Again, unclear why focusing on dominant taxa and not the whole community.

L114 Based on what criteria, these were chosen?

L116 I assume the authors mean "variation in Bray-Curtis dissimilarity index"

L116- L128 Are these surprising or informative? The abundance (and frequency) trends of many taxa are similar in HTS datasets, i.e the abundance and diversity of subsets of OTUs are not independent from others. Do the authors know any dataset that this is not the case?

L120- L128 I am not sure what message the authors try to convey here.

L129- L142 What is the novelty?

L133- L139 Sp. or Spp.?

L170- L171 This has also been reported on local scale and large scales, which need to be cited here.

L197- L198 The greater habitat preference of these taxa may stem from their lower abundance. Did the authors consider that?

L205- L207 This comparison is not relevant, as there are multiple times more bacterial phyla than fungal phyla.

L206- L207 What does it mean?

L209- L210 How about the study by Tedersoo et al. 2014 (Science; 346.6213: 1256688)?

L203- L204 How about dispersal limitation or historical factors?

L236- L248 How did the authors estimated that these groups are highly generalists? Isn't the genus level too coarse for making such a generalization?

L341 What platform was used? How many runs? Did the authors use control samples? More details are needed to evaluate the robustness of this analysis.

L315- L332 This section mostly describes what we already know about fungal community ecology. What is the novel insight gained from this study that can further our understanding of fungal ecology?

Figure 1. How the tree was made, given that it is difficult to align ITS sequences?

Figure 2A&B To me these are different ways of showing the decline of fungal diversity (which is highly correlated to the abundance of dominant fungi, as the authors report) with increasing latitude, as shown previously e.g, in Tedersoo et al. 2014 (Science).

Figure 2C Isn't this graph just another way of showing the clustering in ordination plot? Is it surprising for the authors to see such clustering according to the habit type?

Figure 2E. These maps are highly unreliable, as the authors do not have samples for many areas of the globe. They need to be cross-validated by previous studies.

Figure 3- The results are weak. Non-dominant Ascomycota seem to have the highest potential for being generalists, which together with a number of other pitfalls noted above, cast doubt on the authors conclusions. Was there any correlation between the abundance of taxa and number of their genes?

Reviewers' comments:

Reviewer #1 (Remarks to the Author):

This manuscript present information on the distribution and ecological preference of fungi at a global scale, analysing 235 soil samples collected across the globe.

This is an interesting and appealing manuscript. While I do like the approach and the results presented, I do have a number of comments related to interpretation, analysis and presentation of the results. In particular, the identification of the dominant soil fungi needs further explanation (e.g. why these 83 fungal taxa were selected and not the 33 taxa found in > 50% of the samples (see below for specific comments). This because interpretation of the results possibly depends very much on which taxa are included.

Specific comments:

Please provide additional information on the rationale as to why 83 dominant taxa were selected (e.g. why not 95 taxa). It is important to explain this in detail because the conclusions of this paper depend on the number (and identity) of taxa selection.

Answer: We thank the reviewer for her/his positive comment on the originality and quality of this work, and his/her appreciation of the importance of the findings reported in our manuscript. We have now further clarified our approach in lines 264-268 (Discussion).

“To detect and characterise the globally dominant soil taxa, we selected those fungal phylotypes that were abundant (top 10% most common phylotypes sorted by their percentage of the total ITS rRNA reads), frequent (i.e., occurring in at least one third of the samples from a given biome), and had the highest habitat breadth (i.e., being dominant in at least half of the sampled biomes).”

In this sense, it would help to include a rank-abundance diagram for the 150 (or 200) most abundant taxa (Figure S2b is not so clear because all taxa are included). This shows the reader why these 83 taxa were included (perhaps the 83 taxa identified could also be coloured red in Figure S2a).

Answer: A rank-abundance diagram for the most abundant (top 10%) taxa has been added into the Supplementary material (Fig. S4).

Would it be possible to present a phylogenetic tree (including genetic distance) of the 83 dominant taxa in the Supplement (I wonder how many taxa are genetically highly similar/genotypes of the same species (e.g. very high sequence similarity).

Answer: As requested, a phylogenetic tree of the dominant taxa has been added to the Supplementary Materials (Fig. S11), together with pairwise distances (Supplementary Table S6). The existence of this information has been highlighted in lines 113-125 (Supplementary):

“We further compared the two studies by constructing a phylogenetic tree based on sequences similarities between the dominant phylotypes obtained in the two studies (Figure S11), and estimating the evolutionary divergence between such sequences (Supplementary Table S6). The phylogenetic relationship was inferred by using the Maximum Likelihood method based on the General Time Reversible model⁶. Initial tree(s) for the heuristic search were obtained automatically by applying Neighbor-Join and BioNJ algorithms to a matrix of pairwise distances estimated using the Maximum Composite Likelihood (MCL) approach, and then selecting the topology with superior log likelihood value. All positions containing gaps and missing data were eliminated. Evolutionary analyses were conducted in MEGA7⁷. We found that, overall, phylotypes within Sordariales and Pleosporales tended to have low genetic distance (substitution rate <0.2), as shown in the Supplementary Table S6. However, this comparison was based on a small portion (180bp) obtained from the ITS2 region, which might mask further genetic differences among those taxa.”

Line 129-142 presents a discussion on the identity of the dominant fungi. The primer choice has a large impact on fungal community composition (e.g. some “dominant” fungi may not be detected with the ITS2 primer used). Please add a qualifier to inform the reader about this issue.

Answer: We agree with the reviewer on the issues related to the primer choice, and have address this in the main text (Discussion, L281-284):

“However, the diversity of the dominant taxa identified in this study is limited to the fungal taxa amplified with the primer pair used here, and future developments in sequencing technologies or primer design will be critical to identify more of the common members of the soil mycobiome.”

Line 140 states that mainly filamentous fungi were detected. What about yeasts?

Answer: A comment on yeasts has been incorporated (Discussion, L285-288):

“Apart from two phylotypes associated with the genera of yeast fungi *Vishniacozyma* and *Saitozyma*, respectively, most of the dominant taxa retrieved here belonged to culturable genera of free-living filamentous fungi. This suggests that functional traits associated with this group may play an important role in defining dominance relationships³⁰.”

As far as I know Results and Discussion are separate sections for Nature Communications.

Answer: We thank the reviewer for highlighting this issue. Accordingly, the manuscript has been restructured to meet the journal formatting requirements.

Line 166 refers to three global survey of soil fungi (ref 8, 20, 21). Probably it is good to explicitly state what makes this study different from those three studies (e.g. this is nicely explained in the Supplement on page 13 and possibly there are other arguments as well).

Answer: We thank the reviewer for the suggestion. Differences between this and other global studies have been added to the body of text (Discussion, L250-263):

“Surveying soil fungi across wide spatial scales poses a series of methodological and technical challenges. The accurate estimation of a taxon occurrence on a global scale requires to analyse a wide array of biomes, climates, and continents across the globe. Our sampling campaign covered the nine most common terrestrial biomes of the world, surveyed across 235 sites, from 18 countries and 6 continents, making the present global database one of the most inclusive for fungi (cfr. ¹⁴). Additionally, as fungal communities are vastly diverse, adequate sampling depth is a critical requirement to obtain an accurate evaluation of fungal community composition. In this study, we were able to obtain a total of 12M reads, and an average of 47,207 reads/565 phylotypes per sample, resulting in mostly saturated species rarefaction and accumulation curves (Fig. S2A,B), which is indicative of a satisfactory representation of the most common members these communities harbour. Thus, although we are certainly underestimating the actual diversity of soil-inhabiting fungi, we argue that the comprehensive sampling effort and methodology used here are sufficient for a reliable identification of the most common members of the soil mycobiome.”

Line 170 states that soil properties (e.g. soil pH) were poor predictors of the relative abundance of dominant fungal taxa. Is there also no effect of soil pH (and soil properties) on fungal community composition? Several studies do indicate that soil pH is a major driver of fungal community composition. Any explanation for the absence of any clear effect?

Answer: We thank the reviewer for this suggestion. Note that climatic variables, rather than soil properties, are often reported as the most important factors predicting the community composition of fungi (Tedersoo et al. 2014). Soil pH played a minor role here. We have now clarified this important point in lines 296-306 of the Discussion:

“The dominant fungi identified in this study showed clear environmental preferences, and were associated with three different biomes (mesic, forests, and drylands). Interestingly, unlike dominant bacteria²⁶, soil properties (e.g., soil pH) were poor predictors of the relative abundance of dominant fungal taxa. This result is consistent with other global studies reporting a significant correlation of pH and soil elements with particular fungal functional groups (e.g., mycorrhizal fungi ^{14,32-34}), and a generally minor influence of edaphic characteristics on other fungi ^{14,35}. In fact, climate is often reported as the most important environmental factor predicting fungal community composition^{14,16}. Such differences in habitat preferences for dominant fungi confirm the importance of vegetation and climatic parameters (i.e., aridity index, precipitation / evapotranspiration) in determining the composition and community assembly dynamics of soil fungi^{14,35,36}.”

Line 211: phylotype richness was highest in the Ascomycota: I would be good to formally show that the Ascomycota still are the most abundant phylum if there is a correction for the number of Ascomycota fungal taxa? Also is there any evidence that the primer used has higher specificity for the Ascomycota (if not, it would be good to state that).

Answer: The sentence relative to the Ascomycota richness has been removed from the Discussion. Regarding possible biases in the primer pair used here, Ihrmark et al. (FEMS Microbiology Ecology, (2012) 82(3), pp.666-677) reported mismatches with members of the genera *Cantharellus* and *Tulasnella*, which have rapidly evolving rDNA sequences. However, we are not aware of any other bias towards other basidiomycetes. For other important taxa. such as zygomycetes, members of this lineage have been previously retrieved in studies that used the same primers and technologies (see Nallanchakravarthula et al. (2014)

PLoS One 9, e111455; Gkarmiri, K. et al. (2017) Appl. Environ. Microbiol. 83, AEM.01938-17; Hamonts, K. et al. (2018) Environ. Microbiol. 20, 124–140), suggesting that the bias towards this fungal lineage is limited. However, we have highlighted this possible limitation in the supplementary material (see Dominant fungal phylotypes cross-validation, L106-109):

“Members of Mortierellales, including *Mortierella* spp., have been previously retrieved in studies that used the same primer pairs and technologies³⁻⁵, suggesting that the primers used in our survey are not biased towards this fungal lineage.”

Line 265: resistance to antibiotics: Bahran et al. 2018 (Nature) also found effects of antibiotics if I am right. Please verify and discuss this.

Answer: We thank the reviewer for this suggestion. Discussion relative to the production of antibiotics has been included in the main text as follows (Discussion, L368-371):

“Interestingly, antibiotic production has been identified as a global determinant of inter-kingdom (i.e., fungi-bacteria) biotic interactions in topsoil⁴. Our results hint at the possibility that this trait might also confer competitive advantages at the intra-kingdom level, although further studies are needed to corroborate this hypothesis.”

Line 319-325: The future outlook to use manipulative experiments is an interesting and relevant one. What is the role of dominant soil fungi in structuring fungal community composition and fungi-fungi interactions? Perhaps this can be discussed as well.

Answer: Thanks. This is a great point. Our own results suggest that the composition of dominant fungi can largely influence the composition of other fungal taxa (Fig. S3). We have now added some discussion of this important point in lines L307-325 and L396-403 of the Discussion.

L307-325: “Individually, the co-occurrence patterns of the dominant taxa showed no obvious association between coarse-level (i.e., class to family) taxonomic identity and habitat preferences, with most of the orders/families being represented in each main ecological cluster. However, at the genus level, the dominant phylotypes exhibited clear differences in ecosystem preferences, mainly ascribed to differences in climatic conditions. Such result supports the idea that, at least at a global scale, fungal communities might be more vulnerable to climatic changes than bacterial communities, which are often associated to soil properties³⁷. This implies that climatic changes and increasing landscape fragmentation could result in changes to dominance hierarchies in fungal communities³⁸. Our results also suggest that the relative contribution of dominant soil fungi to ecosystem services may change in contrasting regions of the globe, as indicated by their preferential distribution. These findings provide interesting opportunities for testing hypotheses on structure–function relationships in natural soil communities. For example, they could be used to set up manipulative experiments (such as coalescence experiments, or enrichment/removal manipulations reproducing particular sequences of species addition/loss) targeting soils wherein these taxa vary in their abundance in different regions. These experiments would significantly increase our understanding of the relative importance and ecological role of this prominent group of soil-dwelling fungi in contrasting environments, allowing for better predictions on climate-mediated biodiversity and functionality shifts.”

L396-403: “Finally, given the strong relationship between soil fungal communities and ecosystem functions⁶⁰, and the correlation between dominant species and β -diversity of whole fungal communities, we posit that shifts in dominant fungal taxa should be taken into account when predicting changes in ecosystem functions under climate change and increasing habitat fragmentation. Taken together, our findings provide a baseline understanding of dominant fungal identity, distribution, and ecological attributes in global soils. This understanding is critical if we are to develop approaches and strategies aimed at preserving soil microbial diversity and functionality worldwide.”

It would also be interesting to know whether some of the dominant soil fungi can be considered as keystone taxa, or at least discuss this issue (see review by Banerjee et al. 2018, Nature Reviews Microbiology about keystones).

Answer: We thank the reviewer for this interesting suggestion. We indeed looked at possible keystones among the dominant phylotypes, and preliminary analyses suggest that some of these important taxa harbour above-average centrality measures: that is, those taxa likely act as hubs across different ecosystems (see table below).

[Redacted]

Yet, upon further reflection, we decided not to include this analysis in the current manuscript, as we believe that introducing the keystone concept might be confounding in the context of our study, which already contains quite a bit of information. We, however, acknowledge this comment (it is a great idea!) and will work on a separate study to address such an important question in the future.

Line 346: “the and the”???

Answer: We thank the reviewer for noticing this typo. The sentence should read: “Phylotype identification was obtained against the UNITE fungal database.”

Line 345-350: Please provide further information on the data quality and analysis (e.g. sequencing depth per sample; rarefaction curves per sample)

Answer: Rarefaction curves per sample have been added into the supplementary material (Fig. S2), and information on sequencing depth per sample has been incorporated into the discussion as follows (Discussion, L257-260):

“In this study, the use of high-throughput sequencing approaches allowed us to analyse an average of 47,207 reads/564 phylotypes per sample, resulting in mostly saturated sampling and accumulation curves (Fig. S2 A,B), which is indicative of a satisfactory representation of the most common members these communities harbour.”

Line 373: “33 OTUs being found in > 50% of the samples”: Are the conclusions of this study different if those 33 OTUs are taken instead of the reported 83 OTUs.

Answer: Conclusions would be the same, showing similar predominance of ascomycetes and contrasting ecological preferences. However, rather than focusing on the top 30% of the dominant OTUs, here we preferred a more inclusive approach and considering all OTUs meeting our inclusion criteria.

Line 385: “co-occurrence networks”: Is it worthwhile to search for keystone taxa (OTUs)? This might reveal some further interesting patterns.

Answer: As discussed above, we decided not to include keystones in this work.

Line 468: year of the reference is missing.

Answer: Year of the reference has been added, thanks for spotting this typo!

Please provide a map with sample locations (in the supplement)

Answer: A sampling location map has been added in the supplementary material (Fig. S1).

Reviewer #2 (Remarks to the Author):

Egidi et al. report their analysis on the global spatial distribution of the dominant soil fungi using ITS rRNA genes. The analysis and questions in this study are very similar to what the authors published recently on bacteria (Science 359: 320-325, 2018). Similarly, to their previous paper, the authors focused on a number of dominant fungal taxa found in their soil samples.

Although I appreciate the usefulness of such large-scale surveys, I believe that the value of this study resides mostly in the dataset generated rather than the reported patterns, as most findings have been reported previously both on local and global scales. The authors have attempted to highlight the novel aspects of their study in several places in the manuscript, but to me the novelty is limited to the data analysis and presentation and not the findings. For example, network analysis reveals the co-abundance behavior of fungal taxa in different habitats, and correlations simply reflect variation in the abundance of certain soil fungi across different habitats. Such findings are not novel, as any comparison of distinct microbial habitats would reveal such trends.

Answer: We appreciate the opinion of this reviewer, and understand her/his concerns. We agree that our understanding about the environmental factors associated with community composition and diversity of fungi is increasing. However, we know virtually nothing about the dominant taxa of fungi dominating soils across the globe, which was the primary motivation for conducting this study. Previous analyses, restricted mainly to culturable taxa, or based on coarse taxonomic-level comparisons and limited biome surveys, could not address such a question. As such, we believe that our study makes a novel and important contribution to our understanding of the global distribution of soil fungi, which was only possible given the global extent of our database and all the variables we measured.

Our study provides a number of novel discoveries, which we believe represent important advances in the field. We demonstrate, for the first time, that a limited number of ascomycotous fungi, together with even fewer yeasts, can colonise different ecosystems and dominate soils globally. More importantly, we identify which fungi are ubiquitous across the world's soils, characterise their ecological preferences and functional potential, and describe how their distribution changes across different regions of the world. Several of these prominent fungi match with fungi of agricultural and medical importance, further broadening the relevance of the analyses conducted here. Finally, we provide a novel global atlas for the distribution of fungal taxa; such information has been available for plants and animals for centuries, but until now has been completely missing for soil fungi. We envisage this novel information will be of great utility for a wide range of researchers, including those working in soil biology, conservation biology, ecosystem modelling, and land management.

I have also concerns about the generalization of the findings reported by the authors, most importantly due to geographic limits in samples and a strong spatial dependence between samples. In spite of sampling 235 sites, vast global areas were not accounted for (e.g., tropical Africa and Asia) and most sampled sites are spatially aggregated, i.e. covering the same plant community (mostly steppe or shrubland). These samples are thus highly spatially autocorrelated. As these are unlikely to be independent, conclusions derived from them are

thus prone to a type I error. The multicollinearity of space and environment could have been minimized by a more appropriate sampling design.

Answer: This is an important point. As the reviewer can imagine, surveys across such broad spatial scales as done here are burdened with logistical limitations. Yet, we believe that our global dataset is one of the most inclusive existing today, and includes a comprehensive range of biomes, ecosystems and soil types. Importantly, compared with previous studies, which missed major biomes of our planet, such as drylands (Tedersoo et al. 2014), we considered the nine most common biomes of the globe.

Spatial autocorrelation is not a major issue in this study. First, our criterion to identify dominant taxa is shielded against spatial autocorrelation issues as we are looking for taxa present in more than half of the biomes, therefore grouping locations close to each other (see lines 257-261). Second, rather than predicting geographical turnover rates or latitudinal patterns, for which spatial autocorrelation can be an issue, we were interested in identifying the ecological preferences of dominant taxa globally. Addressing this important question required the inclusion of a wide range of environmental properties, such as temperature (-35°C - 37°C), seasonal precipitation (7-127 mm), pH (4-9), and soil C (1.14-34%). Additionally, in order to exclude possible confounding effects due to spatial autocorrelation of environmental variables, we used a partial correlation analysis between those main ecological factors and phylotype relative abundance, while controlling for space (latitude and longitude). Such an approach allowed us to account for spatial autocorrelation when evaluating the discrete influence of environmental variation on the dominant community, and returned significant associations (e.g., aridity: Mesic: $r = -0.47$; $P < 0.001$; Dry: $r = 0.39$; $P < 0.001$). Because of this, we do not believe spatial autocorrelation, which we acknowledge is an important issue when conducting global surveys like ours, is affecting our results or our interpretations. We reported these additional analyses in the Material and Methods (L477-481) and Results (L177-179) as follows:

L477-481: “In order to exclude possible confounding effects due to spatial autocorrelation of environmental variables, we additionally repeated the correlation analysis between the main ecological factors and phylotype relative abundance, while controlling for space (latitude and longitude).”

L177-179: “Significant associations (e.g., Aridity Index: Mesic: $r = -0.47$; $P < 0.001$; Dry: $r = 0.39$; $P < 0.001$) were also obtained when controlling for spatial autocorrelation (i.e., using latitude and longitude as controlling matrix).”

In addition, the authors found only 10 dominant OTUs shared between the two datasets, which shows that dominant taxa uncovered in such datasets depend strongly on methodology or sampling coverage. Was there any correlation between the abundance of the shared dominant OTUs in the two datasets? Please also report the % identity of the shared OTUs. The results of the cross-validation should be reported in the main text.

Answer: We agree with the reviewer that differences in the dominant taxa are likely due to differences in biome coverage and sequencing technology between the two datasets. Notably, while we recognise that the work by Tedersoo et al. (2014) makes an important contribution to our understanding of the global distribution of soil fungi, it also suffers from important limitations in terms of the range of ecosystems sampled and sequencing depth. Indeed, that study had a poor representation of drylands (<1% of the 350 surveyed sites), despite drylands occupying ~45% of terrestrial ecosystems and being one of the main terrestrial biomes.

These differences (highlighted in Maestre et al. 2015, PNAS) might explain the low overlap between the two databases in terms of dominant taxa, and is especially important given the prominent role of aridity in predicting the distribution of dominant fungi globally. In addition, due to the implemented technology (pyrosequencing) in Tedersoo et al. (2014), their data have a much lower sequencing depth than our study (average of 1,951 reads/356 OTUs per sample vs 47,207 reads/565 OTUs per sample in our study), further constraining any significant overlap between the two datasets.

Even with these limitations, Tedersoo et al. (2014) is still the best existing independent database to compare our results with. Therefore, and to broaden the comparison, we have now constructed a phylogenetic tree using the dominant sequences obtained from the two datasets (Figure S11) and calculated pairwise distances (Supplementary Table S6), where we show that sequences assigned to the same genus overall harbour low genetic distance (nucleotide substitution rate <0.2) – that is, based on the ITS2 sequence similarity, they likely belong to closely related species.

Although potentially interesting, the results based on the genomic data are mostly unreliable. First many of the dominant taxa studied are not included in this analysis and the conclusions are driven based on a few available genomes, which make many of the conclusions biased. For example, the abundant taxa that are easily cultured have representative genomes, and they are most likely to be generalists as most genomic data come from non-soil fungi. In addition, even 3% dissimilarity in rRNA may indicate dissimilar genomic content, as fungal genomes show interspecies variability. Such analysis also oversimplifies the geographic trends in genomic potential and expression (e.g. Ellison, Christopher E., et al. Proceedings of the National Academy of Sciences (2011): 201014971; Branco, Sara, et al. Molecular ecology 26.7 (2017): 2063-2076.). And could the authors explain why they only compared Ascomycota and Basidiomycota? What about other fungal phyla? For example, Zygomycota which seem to be rare in this study.

Answer: We agree with the reviewer that these results were limited by the publicly available genomes for dominant fungi at the time of the writing of this manuscript. We were aware of this issue when writing our manuscript and, for this reason, were very careful when drawing our conclusions. We have further underlined this important point in the new version of our manuscript. Nonetheless, our analysis represents the most up-to-date and state-of-the-art comparative analysis of fungal genomes found to be globally dominant from our data set.

Regarding the choice of what genome to include, we focused on the most dominant phyla found in global studies (Zygomycota is not among them; See Tedersoo et al. 2014; Maestre et al. 2015). Thus, the choice of basidiomycetes compared with other phyla was based purely on the fact that this phylum was the second-most represented lineage within our analyses, and there were a sufficient number of available genomes for comparison. We are aware that many of these genomes belong to generalists, which is critical because our dominant taxa were mostly generalist taxa. More importantly, only one single publicly available genome from zygomycetes matched our data, precluding any comparison between dominant taxa within this group.

Finally, we agree with the reviewer that the choice of 97% identity rather than 100% when choosing which genomes to include may indicate dissimilar genomic content between the reference genome and the actual fungal individuals found within our samples, as fungal genomes have been shown inter-species variability. Yet, given the low number of existing

genomes today, 97% similarity is a realistic threshold to infer functional information for fungal taxa. This is just the first logical step to learn more about dominant fungi, but further work over the coming years will undoubtedly provide critical insights into this matter.

The important issues discussed here have been clarified in the main text, and the suggested references included in the discussion (L371-395):

“Overall, the marked variation in genomic potential between dominant Ascomycota and non-dominant Basidiomycota (Fig. 3, Fig. S5) suggests that ascomycetes may be better equipped to withstand environmental stresses and are able to utilise a higher number of resources, leading to more generalist strategies that may contribute to their increased dominance in soils. It will be of interest, as further genomes for dominant and non-dominant soil fungi become available, as well as genomes of multiple isolates within each phylotype are produced, to understand if these genomic patterns still hold true.

Interestingly, we did not observe the same stark genomic contrasts among the genomes of dominant Ascomycota and non-dominant Ascomycota. As our comparison was mainly restricted to saprobic fungi, the lack of strong changes in the relative number of genes could possibly indicate that these genomic traits are highly conserved at the intra-phylum level for fungi with similar life-styles⁵⁷. It is also plausible that gene regulation and expression mechanisms, rather than number of genes *per se*, contribute to determine process rate, and thus fitness and adaptability^{58,59}. However, the limited number of published genomes and the scarce information on other genes or class of genes not considered in this study hamper our ability to comprehensively assess the role of functional differences in explaining observed within-phylum dominance patterns. Further analyses including more genomes from the dominant phylotypes will allow us to reveal possible attributes underpinning the wide geographic distribution of the suite of dominant fungi found in our study. Our analyses here provide information on which of these genomes should be included in future sequencing efforts. As several of these dominant soil phylotypes are from culturable taxa of ecological, agricultural, and medical importance, their genome sequencing will not only improve our understanding of which genomic traits are associated with dominance within an ecosystem, but also provide new tools to a wide range of scientific disciplines.”

L84-L85 This overlooks a number of efforts that have addressed the biogeography of fungi on large scales.

Answer: Thanks. This sentence, and the overall introduction, have been reframed to include more of the previous study and clarify the study rationale (see responses to L85 to L103 below).

L85- L86 This sentence is vague. How many, what proportion, at what taxonomic levels?

Answer (L84 to L86): As mentioned above, this section of the introduction has been rewritten and expanded to include references to previous large-scale studies on fungal biogeography, and more accurate information on the taxonomic level we refer to (i.e., species). The introduction (L87-108) now reads:

“Like many other microbes, high-throughput sequencing technologies have significantly impacted our perception of fungal diversity and ecology, contributing to uncover the role of environmental attributes in shaping the richness and distribution of soil fungal communities

at both local and global scales. At large geographical scales, many fungal taxa appear to be limited by a combination of physical barriers, abiotic features, host occurrence, and genetic restrictions on adaptation⁴, and their distribution appears to be restricted to local or regional areas. For example, phylogeographic studies on medically relevant or economically important individual species of putative ubiquitous fungi, such as crop and human pathogens, have often retrieved narrowly distributed cryptic phylogenetic species⁵⁻⁸. More recently, many meta-barcoding efforts consistently reported significant distance-decay relationships and scarce phylotype-level community compositional overlap amongst communities of soil fungi at large scales⁹⁻¹¹.

While accumulating evidence suggests a strong spatial structuring of soil fungal communities across ecological gradients, indications of large-scale dispersal and ability for some fungi to dominate many environments also exist. For instance, some fungal phylotypes (e.g., members of the genera *Cladosporium*, *Toxicocladosporium*, and *Alternaria*) possess potentially widespread distributions, and may also be highly predominant in different ecosystems in terms of relative abundance^{12,13}. Similarly, the connectedness of biogeographic regions of the world by shared fungal phylotypes indicates that a number of fungi can be detected in multiple continents and biomes^{14,15}. Yet, a systematic and comprehensive assessment of diversity, identity, ecology, and distribution of those abundant and ubiquitous soil fungal taxa across the globe is still lacking, mainly due to limitations in the biomes and soil types covered by previous studies^{14,16}.

L93- L94 Unclear how the authors reached this.

Answer: We thank the reviewer for highlighting this. The Introduction and part of the Discussion have been reframed to explain this and other points raised in this revision. See response to L101-103 below for further details.

L97- L100 All these would apply to studying communities as well. Do the authors believe that studying dominant taxa would reveal different results compared to community studies? I believe not.

Answer: We are not saying this is not valid for community composition. Yet, as dominant taxa are the focus of this study, we limited the discussion to that fraction of the community.

L101- L103 Again, unclear why focusing on dominant taxa and not the whole community.

Answer (L93 to L103): As the reviewer noted previously, a growing body of literature has described the ecological predictors of community-level fungal composition and richness, uncovering many of the patterns associated with microbial distribution across environmental gradients.

Here, we focus on major knowledge gaps: What fungal taxa dominate soils globally? What are their ecological preferences? Can we provide global maps for their distribution? As outlined in the response to the general comments above, this information is almost completely missing for many microbial groups, including fungi. This limits our understanding on the broad applicability of ecological principles across wide taxonomic groups.

We clarified the aims and relevance of this work in lines 109-117 of the Introduction, and further reinforced this in the first paragraph of the Discussion (L238-249):

L109-117: “Identifying and characterising these cosmopolitan and abundant fungi represent a major goal in ecology¹⁷. Interactions among dominant taxa are predicted to disproportionately affect community stability and functioning^{18–20}, particularly among natural microbial communities²¹. As such, determining which fungi are dominant in soils, which environmental variables drive their abundance and distribution and which mechanisms underlie their dominance capabilities constitutes a major scientific advance. This knowledge can also help us to develop tools to predict how they may respond to ongoing environmental changes, ultimately leading to management strategies to improve fungi-mediated ecosystem functions and services.”

L238-249: “Understanding why some species display larger geographic distribution than others is a major goal in ecology. Disentangling patterns of dominance can help to elucidate the relationships among organisms and their environment, as well as the forces shaping biodiversity and co-existence dynamics²³. Importantly, efforts to catalogue dominant species among animal and plant communities have supported the development of fundamental unifying ecological principles²⁴, that are the ultimate foundation of prioritisation strategies for conservation and management of biodiversity on a global scale²⁵. Yet, equivalent efforts are rarely applied to microbial communities²⁶. Identifying such patterns of dominance is particularly important for soil-inhabiting fungi, one of the most ecologically important groups of living organisms. In this study, we addressed this knowledge gap by characterising the identity, global distribution and ecology of dominant fungi identified in soil samples collected from across the world (Fig. S1).”

L114 Based on what criteria, these were chosen?

Answer: the rationale for choosing the dominant taxa has been clarified in the main text, as outlined above (Discussion, L264-268).

L116 I assume the authors mean “variation in Bray-Curtis dissimilarity index”

Answer: We thank the reviewer for pointing out this mistake. The sentence has been amended accordingly.

L116- L128 Are these surprising or informative? The abundance (and frequency) trends of many taxa are similar in HTS datasets, i.e the abundance and diversity of subsets of OTUs are not independent from others. Do the authors know any dataset that this is not the case?

Answer: We agree with the reviewer that such patterns have been reported before. Yet, we still believe that abundance-frequency relationships are important to characterise patterns of occurrence of microbial taxa at a global scale. Therefore, we decided to keep this analysis in the supplementary material. We have, however, reframed the way we report this in the body of text (see response to L120- L128 below).

L120- L128 I am not sure what message the authors try to convey here

Answer (L116-128): As a result of the manuscript reformatting requirements, part of this section (L116-120) has been moved to the Results. To clarify the message and incorporate the reviewer's remarks, these sentences have been modified as follows (L269-276):

“Unsurprisingly, phylotype abundance and frequency were positively correlated, with relatively few phylotypes being both highly frequent and abundant, and the majority of the retrieved fungal taxa restricted in their distribution. Similar correspondence between abundance and geographical distribution have been recently observed for dominant soil bacteria²⁶, with locally abundant taxa tending to occur in a greater proportion of sites and to have wider geographic distributions. These results suggest that both soil prokaryotic and eukaryotic microbial communities broadly parallel patterns of dominance previously documented in plant and animal communities^{27,28}, with locally abundant taxa tending to occur in a greater proportion of sites and to have wider geographic distribution”

L129- L142 What is the novelty?

Answer: The novelty is that we identified 83 taxa of fungi dominating soils globally. These dominant phylotypes belong to Ascomycota, which contrasts with the finding of Tedersoo et al. (2014), who report that members of Basidiomycota are the dominant fungal taxa within soils. This is likely due to differences in sampling design between the two datasets. In particular, the drylands, which were poorly represented in the work by Tedersoo et al., have been found to be dominated by Ascomycota (see Maestre et al. 2015, PNAS).

For the same reasons as above, part of the discussion has been moved into the Results section (L129-139). We highlighted the novelty of the findings in the Discussion as follows (L285-295):

“Apart from two phylotypes associated with the genera of yeast fungi *Vishniacozyma* and *Saitozyma*, respectively, most of the dominant taxa retrieved here belonged to culturable genera of free-living filamentous fungi, suggesting that functional traits associated with this group may play an important role in defining dominance relationships³⁰. Our results are supported by earlier studies that identified many of these genera amongst the frequently recorded members of the soil mycobiome^{5,31}, with the majority of the most ubiquitous and abundant fungi at the global level being from relatively well characterised fungal lineages. By contrast, the unclassified fungi (d:Fungi), although comprising ~20% of the retrieved phylotypes (Fig. S5A), overall exhibited a narrow distribution (Fig. S6C), suggesting that these poorly characterised phylotypes are not dominant on a global scale, and represent locally abundant, but relatively infrequent members of the soil biosphere.”

L133- L139 Sp. or Spp.?

Answer: We thank the reviewer for noticing this inaccuracy. Accordingly, we now refer to the phylotype taxonomic attributes in terms of genus, rather than species. Also, please note that this section has now been incorporated into the Results (L141-148).

“Two phylotypes matched uncultured organisms. The most ubiquitous fungi found in our dataset were members of the Pezizomycotina, namely Sordariomycetes (including members of the genera *Podospora*, *Chaetomium*, *Fusarium*, and *Trichoderma*), Leotiomycetes (genera

Leohumicola, *Talaromyces*, *Cadophora*) Eurotiomycetes (genera *Penicillium*, *Knufia* and *Exophiala*) and Dothideomycetes (genera *Alternaria*, *Aureobasidium*, *Cladosporium*, *Curvularia*), together with two members of the Tremellomycetes (genera *Vishniacozyma* and *Saitozyma*) and one Mucoromycotina (genus *Umbelopsis*) (Supplementary Table S1).”

L170- L171 This has also been reported on local scale and large scales, which need to be cited here.

Answer: References to previous studies have been incorporated as follows (L296-306):

“The dominant fungi identified in this study showed clear environmental preferences, and were associated with three different biomes: mesic, forests, and drylands. Interestingly, unlike observed for dominant bacteria in soils²⁶, soil properties (e.g., soil pH) were poor predictors of the relative abundance of dominant fungal taxa. This result is consistent with other global studies, which reported a significant correlation of pH and soil elements with particular fungal functional groups (e.g., mycorrhizal fungi^{14,32-34}) and a generally minor influence of edaphic characteristics on other fungi^{14,35}. In fact, climate is often reported as the most important environmental factor predicting the composition of soil fungi^{14,16}. Such differences in habitat preferences for dominant fungi confirm the importance of vegetation and climatic parameters (i.e., aridity index, precipitation/evapotranspiration) in determining the composition and community assembly dynamics of soil fungi^{14,35,36}.”

L197- L198 The greater habitat preference of these taxa may stem from their lower abundance. Did the authors consider that?

Answer: We thank the reviewer for suggesting this possible interpretation. However, this part of the discussion has been simplified as follows (L307-311):

“Individually, the co-occurrence patterns of the dominant taxa showed no obvious association between coarse-level (i.e., class to family) taxonomic identity and habitat preferences, with most of the orders/families being represented in each main ecological cluster. However, at the genus level, the dominant phylotypes exhibited clear differences in ecosystem preferences, mainly ascribed to differences in climatic conditions.”

L205- L207 This comparison is not relevant, as there are multiple times more bacterial phyla than fungal phyla.

We agree with the reviewer on this, and modified the sentence as outlined in the response to L206-207 below.

L206- L207 What does it mean?

Answer (L205- L207): We agree with the reviewer on the ambiguity of this and the previous sentence. This part of the discussion has been simplified and now reads (L277-278):

“Most of the identified dominant fungal phylotypes belonged to a single phylum: Ascomycota.”

L209- L210 How about the study by Tedersoo et al. 2014 (Science; 346.6213: 1256688)

Answer: We thank the reviewer for noticing this omission. Reference to the paper of Tedersoo et al. (2014) has been added.

L203- L204 How about dispersal limitation or historical factors?

Answer: We thank the reviewer for suggesting this additional interpretation. However, this part of the discussion has been removed (cfr. Answer to L197-198)

L236- L248 How did the authors estimated that these groups are highly generalists? Isn't the genus level too coarse for making such a generalization?

Answer: We agree with the reviewer on the possible ambiguous use of 'generalist' in this context. Accordingly, the text has been revised, and this part of the discussion now reads as (L347-349):

“We found that many of the dominant phylotypes were associated with taxa of ecological, agricultural, and medical importance, and were characterised by multiple trophic modes.”

L341 What platform was used? How many runs? Did the authors use control samples? More details are needed to evaluate the robustness of this analysis.

Details on how the sequencing was conducted have been added to the Methods as follows (L411-415):

“The extracted DNA samples were frozen and shipped to the Next Generation Genome Sequencing Facility of the University of Western Sydney (Australia). Fungal diversity was determined by sequencing the Internal Transcribed Spacer (ITS) region 2 with primers FITS7/ITS4 68 on a Illumina MiSeq platform (2x300 PE), and both positive and negative controls were included”

Additionally, we have included species rarefaction and accumulation curves (Fig. S2A and B), and a caveat in the discussion informing the reader about the strengths and limitations of the methodological approach followed (L250-263):

“Surveying soil fungi across wide spatial scales poses a series of methodological and technical challenges. The accurate estimation of a taxon occurrence on a global scale requires to analyse a wide array of biomes, climates, and continents across the globe. Our sampling campaign covered the nine most common terrestrial biomes of the world, surveyed across 235 sites, from 18 countries and 6 continents, making the present global database one of the most inclusive for fungi (cfr. ¹⁴). Additionally, as fungal communities are vastly diverse, adequate sampling depth is a critical requirement to obtain an accurate evaluation of fungal community composition. In this study, we were able to obtain a total of 12M reads, and an average of 47,207 reads/565 phylotypes per sample, resulting in mostly saturated species rarefaction and accumulation curves (Fig. S2A,B), which is indicative of a satisfactory

representation of the most common members these communities harbour. Thus, although we are certainly underestimating the actual diversity of soil-inhabiting fungi, we argue that the comprehensive sampling effort and methodology used here are sufficient for a reliable identification of the most common members of the soil mycobiome.”

L315- L332 This section mostly describes what we already know about fungal community ecology. What is the novel insight gained from this study that can further our understanding of fungal ecology?

As remarked above, by comprehensively sampling the nine most common biomes of the globe, our study goes further than previous studies, which focused on patterns inferred mainly from globally distributed mesic (Tedersoo et al. 2014) or arid (Maestre et al. 2015, PNAS) ecosystems. We demonstrate that climate-distribution patterns hold true even if considering just the top 0.1% of the fungal community, and that aridity is the key driver of such patterns on a global scale.

Figure 1. How the tree was made, given that it is difficult to align ITS sequences?

As already specified in the Methods section, to reconstruct the tree we used the matching sequences (99% similarity) obtained from a BLAST search in GenBank (accession numbers are reported in Supplementary Table 1). Most of these representative sequences covered the entire ITS region (i.e., ITS1+5.8S+ITS2), allowing for a satisfactory alignment, and thus tree reconstruction.

Figure 2A&B To me these are different ways of showing the decline of fungal diversity (which is highly correlated to the abundance of dominant fungi, as the authors report) with increasing latitude, as shown previously e.g, in Tedersoo et al. 2014 (Science).

Answer: Tedersoo et al. (2014) investigated shifts in richness with increasing latitude, while Fig. 2A provides evidence that dominant taxa are abundant across all biomes. We believe that these are substantially different results.

Figure 2C Isn't this graph just another way of showing the clustering in ordination plot? Is it surprising for the authors to see such clustering according to the habit type?

Answer: We thank the reviewer for this suggestion. Yes, we could have visualised the phylotype-associated clusters in an ordination. However, it should be noted that, in ordination plots, partitions/divisions have to be defined *a priori* – that is, ordinations alone cannot be used to find groupings occurring among OTUs. Rather, methods such as the co-occurrence networks conducted here allow to explore such relationships. Based on that, we believe a co-occurrence network is the correct way to report these clusters.

Figure 2E. These maps are highly unreliable, as the authors do not have samples for many areas of the globe. They need to be cross-validated by previous studies.

Answer: We appreciate the comment from this reviewer. We have now conducted three cross-validation analyses for our global maps using ours and independent large scale

databases, and found that our maps are robust. Such cross-validations have been reported in the Supplementary Material as follows (L144-166):

‘We conducted three cross-validations for our predictive maps of the dominant soil fungi distribution. First, we evaluated the correlation between predicted (maps) and observed values for the standardised abundance of dominant taxa of fungi. The two datasets were always positively and significantly correlated using our own global database (Fig. 2; $r = 0.32-0.61$; $P < 0.001$). Subsequently, we re-built the predictive maps using 66% of the data within our database (randomly selected) and the left 33% of data. We found positive significant correlations between predicted (maps) and observed data (33% of data; $n = 78$) for modules #0 (forest; $r = 0.66$; $P < 0.001$; $n = 78$), #1 (drylands; $r = 0.48$; $P < 0.001$; $n = 78$) and #2 (mesic; $r = 0.48$; $P < 0.001$; $n = 78$). Also, the original maps and the new maps were positively correlated, as follows: modules #0 (forest; $r = 0.46$; $P < 0.001$; $n = 225530$), #1 (dry; $r = 0.51$; $P < 0.001$; $n = 225530$) and #2 (mesic; $r = 0.62$; $P < 0.001$; $n = 225530$). Finally, we used the data from the Biomes of Australian Soil Environments (BASE) soil microbial diversity database (Bissett et al. 2016). BASE is a continental-scale dataset for soil microbes covering a wide variety of bioregions, vegetation and land-use types, allowing us to further cross-validate our results using a large-scale, independent database. First, we extracted the phylotypes from BASE matching (sequence similarity $>97\%$) the top dominant taxa found in our global dataset, resulting in 234 phylotypes accounting for, on average, 9.7% of the total reads. We then focused on the two ecological clusters harbouring the largest number of phylotypes matching our dominant taxa (associated with mesic and dryland systems) for these analyses. Predicted (our global maps) and observed values for the standardised abundance of dominant taxa of fungi were positively and significantly correlated ($r = 0.22-0.33$; $P < 0.001$; $\rho = 0.32$; $P < 0.001$), supporting the validity of our global maps.’

Please note that we did not use Tedersoo global database for this cross-validations because it is highly biased towards mesic systems, hampering the validity of such cross-validation (especially considering that aridity was the most important factor associated with dominant taxa in our study).

Figure 3- The results are weak. Non-dominant Ascomycota seem to have the highest potential for being generalists, which together with a number of other pitfalls noted above, cast doubt on the authors conclusions. Was there any correlation between the abundance of taxa and number of their genes?

Answer: As highlighted in the response to general comments above and in the main text (lines 360-395), we found substantial indications of the occurrence of an inter-phylum correlation between dominance and functional potential. Conversely, the analysis at inter-phylum level was limited mainly by the number of accessible genomes. We agree with the reviewer that more data will help to elucidate the relationship between functional potential and geographical distribution at a finer taxonomic resolution. In this sense, the list of dominant fungal taxa provided here will offer a solid indication on which fungi future genomic sequencing efforts should concentrate.

As illustrated in Fig. 3A, there was no correlation between taxon abundance and number of genes.

REVIEWERS' COMMENTS:

Reviewer #1 (Remarks to the Author):

The authors addressed most of my comments. I still have a several comments that need to be addressed:

The primer choice surely also has an influence on the most dominant fungal taxa (I am sure some dominant fungal taxa are not detected because the primer does not detect them or discriminates against them. The needs to be explicitly stated in the discussion.

I realize when looking at the map with sample locations (Figure S1), that only few samples in Northern locations are included (2 in Alaska). However, samples from large countries Canada, Russia and North European countries are missing. Similarly, locations from the tropics (e.g. Brazil, Africa, South-East Asia) are largely missing. The sampling design has a big impact on the "detection" of dominant taxa (e.g. with 50 locations in the Northern countries mentioned above, probably some other dominant taxa (typical for humus rich soils in Taiga/Tundra) would be included. This is something that needs to be very clearly addressed in the discussion (same comment could be made for samples from the tropics). As such the conclusions need to be toned down because with additional sites in those locations missing, I am sure there will be some changes in the dominant fungal taxa detected.

The authors added a rank-abundance diagram for the most abundant (top 10%) taxa into the Supplementary material (Fig. S4). This helps and it also makes a bit clear why 83 taxa were included. However when look at Fig S4b (close –up) it is still unclear what is the cut-off level (please indicate that). I would say, that it would be around 50 (because afterwards, there is hardly any difference (in terms of OT U relative abundance between sample 80 and sample 100)).

As requested, a phylogenetic tree of the dominant taxa has been added to the Supplementary Materials (Fig. S11). Are the sequences of these dominant taxa deposited somewhere (possibly this is mentioned elsewhere). But make a note in the figure legends where those sequences can be found (e.g. the sequences need to be deposited somewhere). Would it also be possible to name the most abundant and most widespread taxa (e.g. OTU1 etc.) in Fig S4a – those taxa that are more or less separate from the large group in the left part of the figure. It is interesting that most taxa in Fig S11 are Ascomycota. Other phyla are missing (e.g. for instance Glomeromycota are missing, possibly because the primer used to analyse the fungal communities does not detect them or discriminates against them. Would it be possible to name a few fungal groups that are expected to be missing (or in low abundance) because of primer choice (e.g. in the discussion (L281-284)?

Answer of the authors to my query: Members of Mortierellales, including *Mortierella* spp., have been previously retrieved in studies that used the same primer pairs and technologies 3–5, suggesting that the primers used in our survey are not biased towards this fungal lineage. Reviewer: Despite this, I feel the primer choice has a huge impact on which fungal sequences are being detected. Even if some members are detected (e.g. such as Mortierellales), the primer can still discriminate against such taxa (or other taxa). I feel it is better to be modest here (and avoid very strong conclusions in the paper) and state that further research is necessary (also in view of sampling issues as mentioned above).

Reviewer #2 (Remarks to the Author):

REVIEWERS' COMMENTS:

Reviewer #1 (Remarks to the Author):

The authors addressed most of my comments. I still have a several comments that need to be addressed:

The primer choice surely also has an influence on the most dominant fungal taxa (I am sure some dominant fungal taxa are not detected because the primer does not detect them or discriminates against them. The needs to be explicitly stated in the discussion.

I realize when looking at the map with sample locations (Figure S1), that only few samples in Northern locations are included (2 in Alaska). However, samples from large countries Canada, Russia and North European countries are missing. Similarly, locations from the tropics (e.g. Brazil, Africa, South-East Asia) are largely missing. The sampling design has a big impact on the “detection” of dominant taxa (e.g. with 50 locations in the Northern countries mentioned above, probably some other dominant taxa (typical for humus rich soils in Taiga/Tundra) would be included. This is something that needs to be very clearly addressed in the discussion (same comment could be made for samples from the tropics). As such the conclusions need to be toned down because with additional sites in those locations missing, I am sure there will be some changes in the dominant fungal taxa detected.

Answer: We agree with the reviewer on the limitation of the current methodologies and sampling coverage. We have clarified the need of further investigations in the discussion as follows (Discussion, L 256-265)

“However, the diversity of the dominant taxa identified in this study is constrained by the fungal taxa amplified with the primer pair used here, and to the biomes included in our survey. For example, we are potentially underestimating the distribution of some fungal lineages that are poorly amplified with commonly used ITS primers, such as members of Glomeromycota and Archaeorhizomycetes³⁰. Importantly, our dataset had limited representation of boreal and tropical systems, possibly limiting the number of dominant fungal taxa characteristics of these biomes. Therefore, we envisage that future studies including under-sampled regions of the world, will allow to identify more of the common members of the soil mycobiome.”

The authors added a rank-abundance diagram for the most abundant (top 10%) taxa into the Supplementary material (Fig. S4). This helps and it also makes a bit clear why 83 taxa were included. However when look at Fig S4b (close –up) it is still unclear what is the cut-off level (please indicate that). I would say, that it would be around 50 (because afterwards, there is hardly any difference (in terms of OT U relative abundance between sample 80 and sample 100).

Answer: As requested, we added a cut-off value for relative abundance of 0.1% in the figure (corresponding to the top 130 phylotypes).

As requested, a phylogenetic tree of the dominant taxa has been added to the Supplementary Materials (Fig. S11). Are the sequences of these dominant taxa deposited somewhere (possibly this is mentioned elsewhere). But make a note in the figure legends where those sequences can be found (e.g. the sequences need to be deposited somewhere). Would it also be possible to name the most abundant and most widespread taxa (e.g. OTU1 etc.) in Fig S4a – those taxa that are more or less separate from the large group in the left part of the figure.

Answer: We thank the reviewer for this suggestion. The sequences relative to the dominant taxa in our dataset are now available in Supplementary Table 1. Information relative to their existence is mentioned in Figure 1 (legend).

It is interesting that most taxa in Fig S11 are Ascomycota. Other phyla are missing (e.g. for instance Glomeromycota are missing, possibly because the primer used to analyse the fungal communities does not detect them or discriminates against them. Would it be possible to name a few fungal groups that are expected to be missing (or in low abundance) because of primer choice (e.g. in the discussion (L281-284)?

Answer: We appreciate this suggestion. Accordingly, we have included examples of possible missing fungi in the discussion as follows (Discussion, L 256-265):

“However, the diversity of the dominant taxa identified in this study is constrained to the fungal taxa amplified with the primer pair used here, and to the biomes included in our survey. For example, we are potentially underestimating the distribution of some fungal lineages that are poorly amplified with commonly used ITS primers, such as members of Glomeromycota and Archaeorhizomycetes³⁰.”

Answer of the authors to my query: Members of Mortierellales, including *Mortierella* spp., have been previously retrieved in studies that used the same primer pairs and technologies 3–5, suggesting that the primers used in our survey are not biased towards this fungal lineage. Reviewer: Despite this, I feel the primer choice has a huge impact on which fungal sequences are being detected. Even if some members are detected (e.g. such as Mortierellales), the primer can still discriminate against such taxa (or other taxa). I feel it is better to be modest here (and avoid very strong conclusions in the paper) and state that further research is necessary (also in view of sampling issues as mentioned above).

Answer: We agree with the reviewer on this important matter. Accordingly, we have reworded this section of the supplementary material as follows:

“While members of Mortierellales, including *Mortierella* spp., have been previously retrieved in studies that used the same primer pairs and technologies^{3–5}, we cannot exclude that the

primers used in our survey are not biased against this fungal lineage.”

The importance of conducting further research has been highlighted in the main text (Discussion, L 256-265).

We have invited Reviewer 1 to comment on these remarks. Although Reviewer 1 agrees with many comments of Reviewer 2, she/he feels the study provides very useful information. In addition to the comments in her/his own report, Reviewer 1 makes the following additional comments/suggestions:

1. It needs to be clearly stated that it is likely that additional dominant soil fungi will be detected if additional sites (tropical forests, taiga/tundra at Northern latitudes, etc.) are included.

Answer: The necessity of further sampling has been clearly stated in the main text (Discussion, L 256-265)

2. The ITS2 region is widely used for environmental sequencing of fungal communities and the authors might add a table to the supplement with studies that used this primer, to show the generality of their approach. That said, the authors should clearly state the results of this study would be different if additional or different primers were used (or if primers/sequencing approaches are used that target a longer read length)

Answer: We thank the reviewer for this suggestion. In order to support our choice to target the ITS2 region in the present meta-barcoding study, we have included a reference to Blaaliid et al. "ITS 1 versus ITS 2 as DNA metabarcodes for fungi." *Molecular ecology resources* 13.2 (2013): 218-224 (Methods, L374). In that study, the direct comparison between ITS1 and ITS2 regions clearly demonstrates that the two meta-barcoding regions yield similar results in terms of taxonomic coverage, validating our choice to use the ITS2 region. Limitations due to primer usage and read length have been incorporated into the discussion, as outlined above (Discussion, L. 234-236)

3. The authors should give information about the read length in their analysis

Answer: Information on sequence length has been added into the Methods (L401-403) as follows:

“All the merged reads had an expected error < 0.5 ; the quality-filtered reads were clustered into operational taxonomic units (OTUs) or phylotypes of the length of 180 bp, at both the 97 and 100% similarity thresholds using UPARSE⁶³ and UNOISE⁶⁴, respectively.”

4. Reference 61 is a bit unusual and should be replaced or complemented by another reference.

Answer: Reference 61 has been changed with Ihrmark, et al. *FEMS microbiology ecology* 82.3 (2012): 666-677.

5. The ITS region is not perfect for making a phylogenetic tree (e.g. the sequence length is variable and not so long). The authors could address this issue by comparing their phylogenetic tree with phylogenetic trees in the literature (e.g. if it deviates substantially they

should consider to remove it).

Answer: Although the tree in Figure S11 had some supported branches, we agree that the ITS marker might be not ideal to infer phylogenetic relationships among such distantly related taxa. Therefore, we decided to remove the tree and associated table (Supplementary Table 6) from the supplementary material altogether.